# Correcting Diacritics and Typos with a ByT5 Transformer Model

**Lukas Stankevičius** [1,*] **, Mantas Lukoševičius** [1,*] **, Jurgita Kapočiūtė-Dzikienė** [2] **, Monika Briedienė** [2] **and Tomas Krilavičius** [2]

[1] Faculty of Informatics, Kaunas University of Technology, LT-51368 Kaunas, Lithuania
[2] Faculty of Informatics, Vytautas Magnus University, LT-44404 Kaunas, Lithuania; jurgita.kapociute-dzikiene@vdu.lt (J.K.-D.); monika.briediene@vdu.lt (M.B.); tomas.krilavicius@vdu.lt (T.K.)
* Correspondence: lukas.stankevicius@ktu.lt (L.S.); mantas.lukosevicius@ktu.lt (M.L.)

**Abstract:** Due to the fast pace of life and online communications and the prevalence of English and the QWERTY keyboard, people tend to forgo using diacritics, make typographical errors (typos) when typing in other languages. Restoring diacritics and correcting spelling is important for proper language use and the disambiguation of texts for both humans and downstream algorithms. However, both of these problems are typically addressed separately: the state-of-the-art diacritics restoration methods do not tolerate other typos, but classical spellcheckers also cannot deal adequately with all the diacritics missing.In this work, we tackle both problems at once by employing the newly-developed universal ByT5 byte-level seq2seq transformer model that requires no language-specific model structures. For a comparison, we perform diacritics restoration on benchmark datasets of 12 languages, with the addition of Lithuanian. The experimental investigation proves that our approach is able to achieve results (>98%) comparable to the previous state-of-the-art, despite being trained less and on fewer data. Our approach is also able to restore diacritics in words not seen during training with >76% accuracy. Our simultaneous diacritics restoration and typos correction approach reaches >94% alpha-word accuracy on the 13 languages. It has no direct competitors and strongly outperforms classical spell-checking or dictionary-based approaches. We also demonstrate all the accuracies to further improve with more training. Taken together, this shows the great real-world application potential of our suggested methods to more data, languages, and error classes.

**Keywords:** natural language processing; diacritics restoration; typo correction; transformer models; ByT5; QWERTY

## 1. Introduction

Since the dawn of the computer era, the English language, Latin alphabet, and the QWERTY keyboard are the "computer-native" means of communication. English remains the lingua franca in IT, science, and many other fields. Many people use it in addition to other, their native languages, as do we here.

Most other languages that use a Latin-based alphabet have some diacritic signs ("č") that are added to the basic Latin characters ("c"), modifying their pronunciation. The initial ASCII character set was greatly expanded by the wide adoption of the Unicode Standard to accommodate for the characters of other languages. Typing these characters, however, is not always convenient.

Many different keyboard layouts exist, they can be more efficient for other languages, as well as English, it is easy to remap physical keyboards in software, and virtual keyboards on touchscreens can even be dynamic; however, learning to type efficiently on different layouts is not easy, they are also not universally available. In addition, large alphabets are not practical to fit on a keyboard layout so that each character can be typed by pressing just one key, instead requiring combinations or sequences of keys.

All these factors made the QWERTY variations (including the similar QWERTZ and AZERTY) remain the most popular keyboard layouts for Latin-alphabet-based languages, where the diacritics are usually an afterthought.

By necessity, haste, or convenience, people often forgo the diacritic signs and special characters in the languages that need them, and type using the base Latin alphabet and keyboard layout instead. Such texts can typically be largely understood nonetheless, but this introduces ambiguities and is not considered a proper use of the language.

Our aim, in this work, is to investigate automatic methods of restoring diacritic signs in such texts, as well as correcting other typical typographic errors, colloquially known as "typos", as such fast, sloppy typing usually results in both.

Restoring diacritics (as well as correcting typos) is important for the human readability of the texts, as well as disambiguation and the proper use of the language (and the prestige associated with it), preventing its degradation.

On the more objectively-measurable technical side, undiacritized texts are also harder to proccess automatically: machine-translate, synthesize, parse, etc. The relevance and importance of diacritics restoration are revealed by evaluating them on the downstream tasks, i.e., extrinsically. There are several examples. The diacritics restoration helped to increase the automatic speech recognition quality for the Romanian language when diacritics were restored in the corpus used for the language model training [1,2]. The diacritics restoration also resulted in a better text-to-speech performance for Romanians [3]. Used as the integrative NLU component, the diacritics restoration also improved the accuracy of the intent classification-based Vietnamese dialogue system [4,5]. Similarly, statistical machine translation performance was positively correlated with correctly diacritized words for Arabic [6]. Moreover, a higher binary classification accuracy was achieved after Turkish text diacritization [7].

Usually, the progress in any Natural Language Processing (NLP) topic initially begins with research for the English language and then spreads to others, but the omitted diacritics problem is an exception. The English written language is highly dependent on the original Latin alphabet. Undiacritized ASCII equivalents of a few English loanwords with diacritics (as "café", "naïve", "façade", etc., mostly borrowed from French) do not cause ambiguity and, therefore, can be easily restored with a dictionary. The level of ambiguity and complexity of restoration for the other languages strongly depends on the language characteristics. For languages where the omitted diacritics cause fewer disambiguation problems, the diacritics restoration is formulated as a spelling correction task. In this research, our focus is on languages that already have lexical and inflective ambiguity. Hence, the omitted diacritics exacerbate this problem even more, and simple solutions are not enough.

Virtually all the previous works (see Section 2) investigated the diacritics restoration problem in isolation, i.e., restoring diacritics in otherwise correct texts. This is, however, not realistic: if not enough care and attention is given to using proper diacritics, while typing a text, then, typically, the same is with using the correct spelling. A carefully-typed text without diacritics might be more common in the past, when Unicode was not widely supported for technical reasons, but this is no longer the case. Crucially, it is neither easy to correct typos before restoring diacritics, as those are not proper texts, nor after, as diacritics would not be restored on mistyped words. If, in addition to the missing diacritics, other typographical errors are introduced (as is common with fast, careless typing), specialized diacritics restoration algorithms break down.

Considering these limitations and trends in the current state of the art in diacritic restoration and typo correction, we take an approach with these main contributions:

- In contrast to the current state of the art, we use the latest universal sequence-to-sequence byte-level transformer model ByT5 [8] that has no task- or language-specific structure, vocabulary, or character set;
- We experimentally investigate the effectiveness of this universal method in restoring diacritics on a standard set of 12 + 1 languages, comparing it to the state of the art;

- We experimentally investigate the effectiveness of this universal method in correcting typos while simultaneously restoring diacritics on the same set of 12 + 1 languages.

The rest of this paper is organized as follows. We provide a review of related work in the literature on diacritics restoration and typo correction in Sections 2 and 3, respectively. In Section 4, we give a detailed background on our chosen approach and related transformer models in general. In Section 5, we describe the datasets used. In Section 6 we outline the experimental setting, and in Section 7, we present the results. Finally, we discuss the findings of this work in Section 8 and summarize them in Section 9.

## 2. Related Work on Diacritics Restoration

Restoring diacritics is important, as most of the world's languages use and often lose them in the digital age, as discussed above. Thus, there are many automatic solutions investigated in scientific literature.

### 2.1. Classical Approaches

The first approaches were based on rules and simple text statistics.

#### 2.1.1. Rule-Based Approaches

The oldest practicable solutions achieving an acceptable accuracy for the diacritics' restoration problem are based on a set of rules. The creation of the rules typically requires human intervention and linguistic skills. They also often employ external language resources, i.e., morphological analyzers, syntactic analyzers, and/or morpho-phonological processing tools [9,10]. The authors in [11] use the lemmatization technique to restore diacritics for the Czech language. Their method contains the set of if-then rules that consider prefixes and suffixes.

As presented in [12], different language resources (i.e., a word-based language tri-gram model with the back-off strategy, augmented with the letter-based language model and the extremely simple morphological model) can be integrated into a cascade of finite-state transducers to restore diacritics on the Arabic texts. Changing diacritics changes not only the syntax, but the semantics of a target (ambiguous) word.

The authors in [13] use a rule-based algorithm to determine the implication of relationships between undiacritized and diacritized words by computing distances and conflicts between them based on a distance-map tuned over a long domain experience. Despite the fact that handcrafted rules are less flexible to include all aspects of the language and are harder to transfer to new domains, they are still in use today (1) when the solving task and domain are very specific; (2) if there is no possibility to get the training corpus of a sufficient size and diversity; and (3) as the baseline approach or for comparison purposes.

#### 2.1.2. Statistics-Based Approaches

In addition to the rule-based approaches, another group that effectively solves the diacritics restoration problems is based on corpus statistics. These methods, in turn, can be further divided into a character-level and a word-level. The word-level approaches are considered to be a more accurate solution, but they typically rely on expensive resources (i.e., monolingual texts to train language models, dictionaries, etc.) that do not cover the non-standard language forms. All of this makes them more language-dependent and, at the same time, less suitable for less-resourced languages. On the other hand, character-level approaches are able to more effectively cope with out-of-vocabulary words and, therefore, can be used to diacritize non-normative language texts (such as posts on social networks, forums, internet comments, etc.) in which the omitted diacritics problem is especially apparent.

The majority of word-level statistical approaches are based on pre-trained probabilistic $n$-gram language models [14]. The $n$-gram language models are trained on large monolingual corpora and give a probability of encountering a particular sequence of $n$ words in a text. The robustness of $n$-gram models directly depend on the size and variety of the

training data. The higher the order $n$ of the $n$-gram model is, the lower perplexity it has, and the better it is at language modeling. Yet, high orders of $n$ require a vast amount of data for training and, as a side effect, inflicts sparseness, which leads to zero conditional probabilities. The models are usually based on the closed-world assumption, where words not found in the language model do not exist. Therefore, smoothing mechanisms become especially important in coping with unseen words (typically assigning non-zero probabilities). Larger $n$s are more cumbersome to store and compute, and are typically less beneficial for languages with a free word order in a sentence; rare combinations make language models very sparse, less robust, and they, therefore, require pruning.

Since longer sequences are less probable, word-level diacritization approaches often allow for back-off or interpolation procedures. The authors of [15] successfully applied their language modeling method to the lowercased Slovak texts. The method compares the surrounding context of the target (undiacritized) word with the related $n$-grams (with $n = 4$). In this way, the method considers three preceding and three following words around the target one. If the 4-gram is not found, the process continues by backing off to trigrams, bigrams, and, if necessary, to unigrams. The whole diacritization process is iterative and sequential: after the diacritized equivalent for some targeting word is determined, the new target is set.

A similar method is applied to the Igbo language [16]. The authors tested the bigram and trigram language models with the back-off strategy and various smoothing techniques, experimentally proving the trigram language model with the Add-1 smoothing to be the most accurate for their diacritization problems.

However, the back-off strategy does not always appear to be the best. An experimentally investigated token bigram language model achieved the highest accuracy on the Spanish texts [17]. It outperformed not only the unigram model, but a bigram language model with the back-off strategy.

The diacritics restoration problem for Spanish is also tackled in [18] and three different methods are investigated. Their first method relies on the Bayesian framework. The idea behind it is that words closer to the target would give more clues about its correct disambiguation and diacritization. The basis of the second method is the Hidden Markov Model (HMM) method, which is able to solve ambiguity problems by indicating different parts of speech. The third method, which is the hybrid of both, is able to overcome the limitations of the Bayesian (which performed poorly on rare words) and the HMM (which relied on the imperfect morphological analysis) models to demonstrate the best performance.

The decision-list approach combines word-form frequencies, morphological information, and collocational information to restore omitted diacritics for Spanish and French languages [19]. First of all, it identifies ambiguity with the help of lexical resources (dictionaries), then it collects the context of $\pm k$ words around the target word. Afterward, it measures collocational distributions (containing the target word) to select the most useful representatives. When the log-likelihood values of these collocations are calculated, the algorithm sorts them into decision lists, performs pruning and interpolation. The prepared decision lists are later used to restore diacritics.

The diacritics restoration system for the Croatian language presented in [20] successfully combines the statistical bigram language model with the dictionary (of 750 000 entries) look-up method. The diacritization process contains three stages. During the first stage, substitution schemes are applied to the raw text result for generating the diacritized candidates; then, the validity of each candidate is determined via a comparison with dictionary forms; and finally, correct forms are selected with the language model. The authors demonstrated the effectiveness of their method not only on the artificial data (newspaper articles that were undiacritized, namely for experiments) but also on the real data (forum posts).

The statistical language model can be created not only on the word level but on the character level, as in [21]. During the first stage, for recognized words, it uses a statistical $n$-gram language model with $n = [1, 4]$ that works on the word level; during the second stage, it processes the out-of-vocabulary words with the statistical $n$-gram character-based

model that works on the character level. The authors proved that their offered approach led to the better diacritization accuracy of the Arabic dialectal texts.

### 2.1.3. Translation-Based Approaches

Sometimes the diacritization problem is formulated as the machine translation problem, but instead of translating from the source language to the target, the undiacritized text is " translated" into the diacritized text. However, such a translation problem is less complex due to a simpler (one-to-one) alignment and decoding.

The phrase-based Statistical Machine Translation (SMT) system has been successfully applied to restore diacritics in the Algiers dialectal texts of the Arabic language [22]. This system uses the Moses (Open Source Toolkit for SMT) engine with the default settings, such as the bidirectional phrase and lexical translation probabilities, the distortion model with seven features, a word and phrase penalty, and a language model.

The SMT-based method was also applied to Hungarian texts [23]. Similar to [22], Moses was used with the default configuration settings (except for the translation model that contained only unigrams, and the language model with $n$ up to 5), monotone decoding, and without the alignment step. However, SMT alone was not enough to solve their task: the agglutinative morphology of the Hungarian language results in plenty of word forms that are unseen by the system with the restricted vocabulary. To handle this, a morphological analyzer was incorporated into the system. It generates candidates for unseen words that are later fed into the Moses decoder. The probability of each candidate was estimated from the corpus with a linear regression model considering its lemma frequency, the number of productively applied compounding, the number of productively applied derivational affixes, and the frequency of the inflectional suffix sequence returned by the analysis.

Despite the problem to be solved in [24] being formulated as a word-to-word translation problem, this is not the typical case with SMT. The authors investigated two approaches that only required monolingual corpora. Their lexicon-based approach (applying the most frequent translation observed from the training data) was outperformed by the corpus-based approach (combining information about the probability of translation and the probability of observing a translation in the given context, via a simple log-linear model). This research is interesting for several reasons. First of all, the effectiveness of the method is proven in several languages, i.e., Croatian, Serbian, and Slovenian. Similarly, the diacritics are restored on both standard and non-standard (Web data) texts. Moreover, the authors also performed cross-lingual experiments by training their model on one language and testing it on another. The cross-lingual experiments revealed that the Croatian and Serbian languages can benefit from each other (training/testing in both directions), whereas the model trained on the Slovenian language was not effective for Croatian or Serbian.

### 2.1.4. Character-Level Approaches

Another important direction in diacritics restoration is character-level approaches. They solve problems that are typically defined as sequence labeling. The iterative process slides through an undiacritized sequence of characters by assigning their diacritized equivalents (labels). Each character is a separate classification instance with the surrounding content as other classification features. Such approaches typically require no additional language tools except for the raw text, which makes them suitable for less-resourced languages. Moreover, character-level methods are robust when dealing with unknown words. Depending on the chosen classifier, this classification process can be viewed as the independent instance-based classification (assuming that each instance is independent) or the sequence classification (considering conditional dependencies between predictions) problems.

The seminal research work in [25] described the instance-based classification technique applied to the Czech, Hungarian, Polish, and Romanian languages. Authors tested different window sizes (of 1, 3, 5, 7, and 9 lower-cased characters to both sides) with two classifiers:

the memory-based approach and the decision tree (C4.5). Their offered method achieved an accuracy which is competitive to word-level approaches.

Another study, presented in [26], described the sequence classification tackled with the MaxEnt classifier. This approach is applied to the Arabic language, but instead of pure character features, it employs character- (character *n*-grams), segment- (words decomposed into prefixes, suffixes, stems, etc.), and part-of-speech tag-based feature types. The successful combination of these diverse sources resulted in a high diacritization accuracy.

Similar to [25], three instance-based classifiers (a decision tree, logistic regression, and the Support Vector Machine, or SVM), with character *n*-grams (from a sliding window) as features, were investigated for the Hungarian language [27]. The decision tree, which is also good at identifying important features and keeping decisions easy to interpret, was determined to be the most accurate. This research is important for several reasons: it claims the effectiveness of the offered approach on non-normative language (web data, Facebook posts) and the superiority over lexicon lookup (retrieving the most common diacritized forms) and hybrid (the lexicon plus character bigrams) approaches in the comparative experiments. However, comparative experiments are not always in favor of character-level approaches.

In [28], the character-level and word-level approaches are compared for the Lithuanian language. The authors used conditional random fields (CRF) as the sequence classifier by applying them to the character-level features. Despite different window sizes (up to 6), the character-based approach was not able to outperform the trigram language model with the back-off strategy. The character-based approach was also not the best choice when applied to the Spanish texts [29]. It was outperformed by the decision list (that combines the simple word-form frequency, morphological regularity, and the collocational information) and the part-of-speech tagging (trained on the tagged corpus with information about the diacritic placement) approaches.

Two approaches, namely, sequence labeling (i.e., sequence classification) and SMT were compared in [30] for the Tunisian language. The sequence classification approach uses CRF as the classifier and is applied to the different character (windows up to 6-grams) and word-level (part-of-speech tags of two neighboring words) features. The SMT approach uses Moses with a 5-gramlanguage model and other parameters set to their default values. The comparative experiments demonstrated the superiority of the sequence labeling approach compared to the SMT approach.

Even more comprehensive comparative experiments are performed in [31], and they cover 100 languages and several approaches, such as the lexicon lookup, the lexicon lookup with the bigram language model, several character-level methods with various window sizes, the hybrid of the lexicon lookup with the bigram language model (for words in the lexicon), and the character-level approach (for words that are not in the lexicon). With some exceptions, the hybrid approach performs the best for the majority of languages.

A similar hybrid approach is also successfully applied to the Romanian language [32]. The candidates for each recognized undiacritized target word are generated based on mappings of the dictionary, and the appropriate candidates are selected with the Hidden Markov Model (HMM)-based language model. The diacritics for unknown words are restored with the character-level approach (described in [25]) using windows with up to eight characters.

Another hybrid approach that is used for completely different purposes (to clarify/claim the output of the character-based method) is presented in [33] for the Turkish language. During the first stage, it performs the sequence classification with the CRF method, but next to current/neighboring character,s it also uses the current/neighboring tokens as features, i.e., five character-level and two word-level features. The output of the first stage is fed into the morphological analyzer-based language validator. The authors compared their hybrid approach with several others (rule-based, rule-based with the unigram language model, and character-based but without language validator stage) and proved it is the best model to use.

In contrast to the previously described approaches, the sequence labeling problem can be solved, not on the character, but the syllable level, as in [34]. The authors solved the instance-based classification problem by treating each syllable as a separate independent classification instance and applying the SVM classifier on top. They used different types of features, such as the *n*-grams of syllables (surrounding the target with window sizes of 2 and 3); syllable types (uppercase, lowercase, number, other), characterizing surrounding syllables, and dictionary-based features (dictionary words that contain the target syllable). The method achieves a high accuracy on Vietnamese texts.

*2.2. Deep-Learning-Based Approaches*

With the era of Deep Neural Networks (DNNs), the diacritics restoration problem is being solved with these innovative techniques. Some of them rely on word embeddings, i.e., learned word representations that are capable of capturing the context.

Word2vec embeddings were integrated into a three-stage diacritics restoration system for Turkish in [7]. During the first stage, candidates are generated for the target word. During the second stage, the morphological analyzer checks if the candidates are legitimate words. During the last stage, the word2vec-based tool evaluates the semantic relationship of each candidate to its neighboring words with the similarity method and chooses the most suitable one. The authors tested two types of word-embedding models (i.e., the continuous bag-of-words model, or CBOW, which predicts the target word based on its context, and the skip-gram model, which predicts the surrounding words based on the input word) and several similarity measures (Cosine, Euclidean, Manhattan, Minkowski, and Chebyshev). Their experimental investigation revealed that the skip-gram and cosinesimilarity approach was the most accurate on Twitter data.

The omitted diacritics problem can also be tackled at the character level and solved as a character classification problem. An example of such a system is for the Arabic language, and the core of it is the Bidirectional Recurrent Neural Network (BiRNN) [35]. The BiLSTM takes the undiacritized character (as an input) and outputs its diacritized equivalent (as a label). The input characters are represented as real-number vectors that are randomly initialized at the beginning and are updated during the training. The output is the *n*-dimensional vector, with the size *n* equal to the size of the output alphabet. The approach outperformed the other methods in the comparative experiments. A similar approach is offered for Hebrew, and the base of it is the two-layer LSTM [36].

The Deep Belief Network (DBN) (as a stack of multiple restricted Boltzmann machines in which each layer communicates with both the previous and subsequent layers; however, the nodes in each layer do not communicate with each other literally), on the character level, is applied to Arabic [37]. The advantage of the DBN compared to the RNN-based approaches is that it overcomes the limitations of backpropagation. The authors tested their approach on several benchmark datasets and compared it to other competing systems, claiming their approach to be the best for the diacritization problem.

The robustness of sequence classification was also tested for Croatian, Serbian, Slovenian, and Czech [38]. However, this language-independent part has the additional integration of the 2, 3, 4, 5-gram language model. This language model-based version, for the inference, uses the left-to-right beam search decoder that combines the neural network and language model likelihoods. The authors compared their method with other approaches (lexicon-based, corpus-based) and systems, demonstrating its superiority over the other models.

The authors in [39] also assumed that pure character information is not enough to achieve a high accuracy for Arabic, because the lexical and syntactic information is closely interrelated. Due to this reason, they offer the multi-task approach, which jointly learns several NLP models, namely for segmentation (operating at the character level), part-of-speech tagging, and syntactic diacritics restoration (operating at the word level). All these aggregated models are later used for diacritics restoration. The segmentation, part-of-speech tagging, and syntactic diacritization models use separate BiLSTM methods with the

softmax on top of each. Their outputs are aggregated, and they become the input for the diacritization model which, again, is BiLSTM-based. The authors compared their model to the other popular approaches, and they claim it is a statistically significant improvement.

A similar character classification problem was solved in [40] for the Romanian language. The architecture of this offered system has three different input paths: for characters (to represent the window of characters around the target character), words, and sentences (in which the target character appears). The character input path is represented by a BiLSTM encoder for character embeddings, the word input path by the FastText word embeddings, and the sentence input path by the BiLSTM encoder applied on concatenated FastText word embeddings. The authors tested their approach with different combinations of input paths (only character input, character input with the word input, etc.) proving that the best accuracy can only be reached with all the three input paths.

The sequence classification tasks were also solved for the Arabic, Vietnamese, and Yoruba languages [41]. The authors tested the Temporal Convolutional Network (TCN) (in which information flows from the past to the future, as in the LSTM) and the Acousal TCN (A-TCN) (where information flows in both directions, as in the BiLSTM) approaches, and compared them to the recurrent sequential models, i.e., the LSTM and the BiLSTM. The A-TCM approach yielded a significant improvement over the TCM and had a competitive performance over the BiLSTM. The hybrid approach (as the three-stage stacked pipeline) for the Arabic language [42] integrates a character classifier as the first language-independent component. The other two components, namely, the character-level deterministic rule-based corrector and the word-level statistical corrector, are already language-dependent, but help to increase the accuracy even further.

Another research direction for the diacritics restoration problem is the sequence-to-sequence (seq2seq) methods. The seq2seq architecture consists of an encoder (converting an input sequence into a context vector) and decoder (reading the context vector to produce an output sequence) blocks as separate DNNs.

Such a seq2seq approach, with the RNN-based core, was successfully applied to the Turkish language [43], and, with the LSTM-based core, to Vietnamese texts [5,44]. In [45], Romanian authors investigated four different encoder-decoder architectures operating on the character level: one-layer LSTMs, two types of stacked LSTMs, and the CNN-based method (three-layer CNN with the concatenated output of the encoder and decoder, processed with another two-layer CNN), and determined that the CNN-based approach was the most accurate. Moreover, they compared their seq2seq approaches with the classification-based approach. The first approach is a hybrid of the BiLSTM (operating on the word level) and the CNN (operating on the character level); the second is described in [38] and requires additional language resources (a language model). The comparative experiments revealed the superiority of seq2seq methods.

Transformer-Based Approaches

The state-of-the-art techniques in the diacritics restoration, as in all NLP fields, employ transformer-based models.

The multilingual BERT was successfully applied to 12 languages (Vietnamese, Romanian, Latvian, Czech, Polish, Slovak, Irish, Hungarian, French, Turkish, Spanish, and Croatian) [46]. The BERT embeddings, created on the undiacritized text, are fed into a fully connected Feed-Forward Neural Network (FFNN). The output of such a network is a set of instructions (as labels) that define the diacritization operation necessary for each character of the input token. The authors claim that their BERT-based approach outperforms all previous state-of-the-art models.

The authors in [47] solve the character classification problem for the Vietnamese language by offering a novel Transformer Decoder method with the Penalty layer (TDP). The model is a stack of six decoder blocks. The encoder part is redundant since each input character corresponds to only one output character. The penalty layer restricts the output by only allowing the possible characters for each input character. The authors also

performed comparative experiments, proving their approach is superior to those offered in [38].

Another transformer-based technique was applied to 14 languages (Bosnian, Czech, Estonian, Croatian, Hungarian, Lithuanian, Latvian, Polish, Romanian, Slovak, Slovenian, Albanian, Serbian, and Montenegro) [48]. The core of the diacritization approach is the Marian Neural Machine Translation (NMT) system [49] with six encoder-decoder layers, which is applied to the frequently occurring character sequences. The research is especially interesting because it is performed in monolingual (training and testing on the same language) and multilingual (by either mixing the data of all languages or by mixing the data of all languages, but inserting language codes as the first token of each segment) settings. The authors experimentally determined that the monolingual experiments gave almost the same accuracy as the multilingual experiments with the language codes.

## 3. Related Work on Correcting Typographical Errors

A typographical mistake is an error that occurs while printing the material. Historically, this was due to errors in the setup of the manual type-setting. The term includes errors caused by mechanical failure or the slipping of the arm (or finger), but does not include errors caused by ignorance, such as spelling errors. However, typos are the subset of a bigger category of misspelling errors. These are of the same importance and are solved with the same methods. The only difference is that typographical errors are easier to model, as they depend only on the keyboard (we discuss it more in Section 5.2) and not the language.

The most classical spelling error correction systems follow these steps:

1. Error detection;
2. Candidate generation;
3. Error correction.

We will cover separate methods constituting this pipeline below.

### 3.1. Non-Word Detection

The dictionary is the most popular error detection method, sometimes called a lexicon or a unigram language model. The dictionary detects non-words, that is, the ones that cannot be found in it. The first system [50] used exactly this method with some additional heuristics. Modern spell checkers, such as GNU Aspell [51] and Hunspell [52] also compare each word of a text to their large lists of words. In Hunspell's case, the dictionary is compacted by keeping only the main word forms with transformation rules, prefixes, and suffixes, thus supporting many languages with rich morphologies.

There are some downsides to the dictionary method. As noted in [53], about 40% of spelling errors are real-word errors (i.e., "from" → "form") and cannot be detected by the dictionary. The study by [54] showed that GNU Aspell corrects only 51% of errors and performs best on non-word errors. Secondly, the dictionary cannot cover rare words, such as proper names, country and region names, technical terms, and acronyms. This issue could be dealt with by enlarging the dictionary. However, [53] argues that, eventually, most of the misspellings would match rare words and would, therefore, fail to be spotted.

### 3.2. Candidate Generation

This is the task of finding the confusion set of real words for a given misspelled word. One can manually craft a confusion set or look for a publicly available one, such as [55] for the Chinese language. However, usually these sets are generated on the fly. The similarity measure between words is obtained by the phonetic or the Minimum Edit Distance algorithms.

The most-known phonetic algorithm is Soundex [56,57]. The cornerstone of the Soundex approach is that homophones (the same-sounding words) are encoded similarly, so that they can be matched regardless of subtle differences in their spelling. A Soundex code is computed from a misspelling, and words that have the same code are retrieved from the dictionary as correction candidates. A similar principle of misspelling encoding

was used in the first system by [50]. Nowadays, the Metaphone representations of words (as an improvement over Soundex) [58] are used in Aspell [51].

The Minimum Edit Distance [59] measure is defined by the minimum number of edit operations needed to transform one string to another. As reported in [60], more than 80% of errors differ from the correct word by only a single letter; thus, the distance between them is low. There are several different edit distance algorithms: Levenshtein [61] (number of insertions, deletions, and substitutions), Damerau–Levenshtein [60] (treating transposition as a single edit), Hamming [62] (number of characters that differ between two equal-length strings), and the Longest Common Subsequence [63]. As an example, the widely-used Aspell uses the Damerau–Levenshtein distance between Metaphone representations of words.

### 3.3. Using Context and External Datasets

The given candidates can be simply ranked by their pre-computed distances. On the other hand, some additional information, whether from nearby words or from additional corpa, can aid target word selection.

The approach in [64] uses a Bayesian combination rule to rank the given candidates. First, the probabilities for substitutions, insertions, and other errors are collected from a corpus of millions of words of typewritten text. Then, given a misspelled word, its each inflection and the resulting word probabilites are combined to produce a probability estimate for each correction candidate.

The $n$-gram language models [14] that are trained on a large external corpus can give a conditional probability of how likely a sequence of words is to be followed by a certain word. The $n$-gram model ranking for confusion sets is used in multiple works for spelling correction [54,65–69]. The character-level $n$-gram also allows for the calculation of a distance measure (such as Hamming in [70]) by comparing the character $n$-grams between two strings [71]. Spelling correction systems using $n$-grams usually employ back-off techniques [65,66,68] or other [72,73] smoothing techniques, and sometimes, due to its size, they even require a complex distributed setting [68,74]. The extensions and problems with the $n$-gram models have already been discussed in Section 2.1.2.

External datasets are especially well-exploited by the neural network approaches. The authors of [75,76] used a FastText [77] shallow neural model to learn both known and unknown word vectors as a sum of character $n$-gram embeddings. Candidate words could then be scored with a cosine similarity to the context words vectors. The differences between these two works are text domains. In the study by [75], the model was trained in the Bangla language, while in the study by [76], the model was trained on English and Dutch clinical texts.

The ability to learn from vast text resources eventually culminated in the state-of-the-art transformer models, discussed in Sections 3.5 and 4.2.

### 3.4. Real-Word Errors

We already reviewed techniques for detecting and correcting non-word typos. The other, far more difficult, group is the real-word errors. These are misspellings that result in other real words. Ironically, these errors are also caused by automatic spelling correction systems [78]. As it is harder to apply unsupervised methods such as the dictionary, there is also a challenge to build tools for different languages with different alphabets and rules [79].

The detection of real-word errors can be done by searching every word in a confusion set and checking for a better alternative [66,72,80,81]. The candidate population is usually done by the $n$-gram method, and others, as already discussed in Section 3.2. Some works employ natural language parsers that check grammar [82,83] or look for words semantically unrelated to their context, that have semantically-related spelling alternatives [84]. Since the detection is similar to the selection of candidates here, the real-word error correction systems often do detection and correction at the same time.

*3.5. Transformer Models for Spelling Error Correction*

Recent advances in natural language processing, particularly the transformer architecture [85], solve many problems encountered in traditional approaches. Firstly, the traditional detect-suggest-select pipeline is discarded. Whether it is a seq2seq translation or an encoder-type each-token classification, target words are generated immediately. Secondly, the segregation of non-word and real-word methods is gone here. Finally, the use of the context from the whole input sequence and the knowledge from the additional datasets are now employed. Despite the advantages, some open issues are still being solved.

An important problem for seq2seq models is the over-correction, which is the attempts of a model to correct the sentence even if it is not confident. The authors of [86] addressed this problem for their Korean spelling error correction system by using a dedicated Copy Mechanism. Correction is attempted only if it detects that the input is incorrect, otherwise, the input sequence is copied. The results showed that such a mechanism resulted in a better overall performance. The authors of [87] found that the over-correction can be mitigated by allowing the transformer to be trained with unfiltered (containing gibberish samples) inputs. In this way, the model is forced to stick to the initial input, unless there is a high certainty of a typo. There is also an attempt to use an additional error detection classification head in the encoder-type transformer model [88].

Usually, small available datasets are not enough to train transformer models. As a result, most works resort to the artificial spelling error generation. The authors of [87] used the statistics of their private 195 000 sample dataset to generate 94 million examples. The authors of [86] used Grapheme-to-Phoneme and Alphabetical (insertions, deletions, and substitutions) generators, together with 45 711 private samples. The authors of [88] constructed a random rule-based generator covering the most common error categories of the Vietnamese language. Works utilizing the BERT [89] encoder can utilize, or supplement, the default masking [MASK] token. The authors of [90] also used related words from confusion sets, while the authors of [91] replaced them with phonologically and visually similar ones.

The original BERT [89] transformer model used subword tokenization. As misspellings happen at a character level, it is wise to also incorporate characters or other phonetic features. The authors of [88] used an additional character-level encoder to output character-level vectors. These are concatenated with word embeddings and are used in the final word encoder. For the Chinese language, [91] additionally added phonetic and shape embeddings acquired from separately-trained single-layer GRU [92] networks. Parallel to the character classification, authors also performed pronunciation prediction. Similarly, other works on the Chinese language find it useful to predict not only characters, but also pinyin and radicals, which is a total of three classification heads. In contrast to these approaches, we use the fine-grained model in the first place and we, therefore, can avoid the additional incorporation of character information.

## 4. Our Methodology

The analysis of related work revealed research performed under very different experimental conditions, which makes the results difficult to compare. Different languages have different levels of complexity and ambiguity, and omitting the diacritics or introducing typos exacerbates this problem even more. The training/testing texts cover normative (fiction, periodical, Bible texts) and non-normative (tweets, comments)language types. Investigated approaches are affected by the availability of language resources and the emergence of new methods, and vary from rule-based, traditional machine learning to the most innovative deep learning solutions. There are different evaluation types: extrinsic, which refers to evaluating the downstream tasks, vs. intrinsic, which refers to calculating the percentage of correctly restored words or characters); different evaluation metrics cover word-level and character-level (including all characters or only with diacritics) techniques. Hence, there is no consensus about which approach is the best for the diacritics restoration and

typographical error correction problems. Recent trends suggest that innovative approaches, such as transformer models, are still needed, and should be the most promising.

### 4.1. Formal Definition of the Solving Task

Let $X = \{x_1, x_2, \ldots, x_N\}$ be a sequence of tokens, constituting our text without diacritics and/or with typos. Let $Y = \{y_1, y_2, \ldots, y_M\}$ be a sequence of equivalents with their diacritics and/or typos corrected. Depending on the chosen tokenization form, a token can represent a word, subword, character, or byte value.

The function $\eta$ correctly maps $X \to Y$. Our task is to find the method $\Gamma$ which is as close to an approximation of $\eta$ as possible.

In this work, we use a transformer model as a method $\Gamma$. Below, we further explain what is behind tokens in our case, and how the sequence mapping is performed.

#### 4.1.1. Tokens

Generally, the text is represented as a Unicode string. It is a sequence of code points, which are numbers from 0 through 1 114 111. For example, the letter "s" has a code point of 115, while the same letter with the additional caron, "š", is at 353. The Unicode describes a huge amount of various symbols but is very wasteful in terms of memory space. The most popular symbols are at the beginning of this list, but they would still have to be represented as 32-bit integers. Instead, UTF-8 encoding is employed to translate the Unicode sequence into 8-bit bytes. If the code point is larger than 127, it is turned into multiple bytes with values between 128 and 255. Therefore, the code point 353 of the letter "š" is translated into two bytes 197 and 161, while the letter "s" retains byte 115. The authors of [8] showed better results using a transformer model ByT5 at these byte-level tokens, rather than on characters. Inspired by their success on transliteration and noisy text tasks, we also use the same byte-level tokenization.

#### 4.1.2. Mapping X to Y

One should note that the transformer model does not map the whole target sequence instantly. Starting with the first artificial start token $y_0$, it estimates the probability for each next token by taking into account the whole input sequence and the previously generated tokens (the context). The probability that the next token is $y_i$ can be written as

$$P(y_i \mid \{x_1, x_2, \ldots, x_N\}, \{y_0, y_1, \ldots, y_{i-1}\}). \tag{1}$$

Thus, the output from a transformer model is a list of probabilities for each token, in a vocabulary, to be the next token $y_i$.

The choice of the next token, given the probabilities of all candidates, depends on the decoding algorithm. There are two groups of maximization-based sampling: greedy and beam search. The most obvious greedy approach is to select a token with the highest probability. During the beam search, a defined number (the so-called beam size) of the word sequences with the highest overall probabilities are kept. This way, a single low-probability word would not shadow a high-overall-probability sequence. Stochastic approaches are inappropriate for our task as there is only one right way to restore diacritics or correct typos.

### 4.2. Transformer Models

There are several key reasons why transformer [85] architecture became the top-performing model in multiple natural language processing leaderboards, such as Super-GLUE [93]. The first reason is that, compared to previous recurrent ones, it is highly parallelizable. It does not need to wait for the calculations to finish for the previous word. Instead, calculations for all words are done at once. Models can be elementary, trained on multiple dedicated machines (such as GPUs), thus quickly digesting vast amounts of data. Secondly, only after a single block (usually called a layer), the information between all tokens is already exchanged. This is accomplished by a self-attention layer inside the block, which processes a sequence by replacing each element with a weighted average of

the rest of the sequence. As there are usually more than five blocks, it allows for the quick learning of long-range dependencies. Finally, it costs less computational power, demanding shorter sequences, which is the case for most of the language tasks. These reasons allowed transformer architecture to flourish.

The capabilities of these models come with a price. Training them from scratch requires dedicated hardware (i.e., a GPU with a large enough memory), takes a long time, and consumes a lot of electricity. Solutions to alleviate this burden started with the introduction of the BERT [89] transformer. This model is pre-trained with a general word-masking task to be fine-tuned for any desired task later (the process called transfer learning). It is estimated that the pre-training of BERT caused more than 300 kg of $CO_2$ emissions [94], but it can be easily fine-tuned for a custom purpose at a small fraction of that cost. Three years later, there are plenty of similarly pre-trained publicly available models (e.g., at HuggingFace transformers library [95]). We also built our work on top of one such pre-trained ByT5 [8] model.

In general, transformer models can be grouped into three categories: auto-encoding, auto-regressive, and sequence-to-sequence. We will cover them in more detail below.

### 4.2.1. Auto-Encoding Transformer Models

This version of the transformer model possesses only an encoder part. It encodes the input text into distinct output vectors for each given token. Attention layers can access all the words in the initial sentence to get the most representative information of the whole sequence. Additional "heads" can be placed on top to further process this representation for a sentence or word classification, extractive question answering, regression, or other tasks. The most popular model of this category is the BERT [89].

Several diacritics restoration works use transformer encoders. The authors of [46] performed a classification of each transformation, described by a diacritic sign to be applied and its position in a word. Meanwhile, the model in [47], although it is named a "decoder", has its attention masking removed and classifies output diacritic mark categories for each input character.

### 4.2.2. Auto-Regressive Transformer Models

These models possess only the decoder side of the original architecture, and its tokens can only attend to the previous ones. Probably the most-known example is one of the latest gigantic (175 billion parameters) transformer models, GPT-3 [96]. It is used in practice by finishing sentence beginnings, which is the so-called zero-shot task solving. In this setting, the human must manage to convey all the necessary information for solving the task in the beginning, such as by providing examples of task solutions. Currently, we do not possess access to the latest GPT-3 model, nor do we believe it can adequately cover the languages we use in this work. However, it would be interesting to test its capabilities in an unsupervised zero-shot multilingual diacritics and typos correction.

### 4.2.3. Sequence-to-Sequence Transformer Models

These are the encoder-decoder models. In the encoder part, each token can attend to every other token. On the decoder side, there are two types of attention that occurs. The first type is the attention to the decoder's past inputs, which is the same as in the auto-regressive transformer models. The second type is the model's full attention to the tokens of the encoder. The most straightforward application of this network is the translation. The encoder only receives input language tokens, while the decoder is fed target language tokens and predicts them one at a time. As the diacritics restoration task can be viewed as a translation task, this transformer type is found in several related works [97–99].

The most popular model of this category is T5 [100]. Authors framed various tasks, even ones including numbers, to text-to-text format. They reported that there was no significant difference if a separate "head" was used, or an answer was generated as simple text. This, in turn, made the model very simple to use. In this work, we use the follow-up

multilingual ByT5 [8] model designed to work with byte-level tokens. We think that the seq2seq approach is the most adequate, as it is universal. Additionally, operating on the byte-level gives a level of immunity to minor text noise, i.e., against typographical errors, and is more language-universal.

### 4.2.4. The ByT5 Model

The ByT5 model [8] is a general-purpose pre-trained multilingual text-to-text model, based on its earlier predecessor, mT5 [101]. It completely disposes of SentencePiece [102] tokenizer, as it does not need any. The authors concentrated 3/4 of the parameters into the encoder by decoupling the depth of the encoder and the decoder. A small version of the ByT5 now has 12 encoder layers and four decoder layers.

In the ByT5 model's case, the total vocabulary size is 384, consisting of: three special tokens (*<pad>* for padding, *</s>* for the end of the sequence, and *<unk>* for unknown), $256 = 2^8$ values of the main eight-bit byte, and 125 extra sentinel tokens used only in the pre-training task. In the small version, the vocabulary accounts only for 0.3% of the total parameters, while in a similarly-sized mT5 model, the vocabulary took 85% of the total parameters. As a result, the small ByT5 model, working with fine-granularity tokens (bytes), outperforms mT5, which worked inefficiently due to its large granularity and its rarely-used vocabulary parts (subwords) which took up much parameter space.

Due to its byte-level nature, the ByT5 model is slower to compute. More fine-grained tokenization produces more tokens for the same text and requires more time for the model to digest. However, the ByT5 model's authors showed that, for short-to-medium length text, the time increase is negligible. This is the case for diacritics restoration, as the input is composed of a single sentence.

The sequence-to-sequence nature of the ByT5 model tackles the limitations of the latest state-of-the-art diacritics restoration model [46], which is based on the BERT. The latter system was an auto-encoding type, and it performed classifications for each token. That is, it had to predict the proper classes of each token correction, described by the position and diacritic sign type. This system is limited to its predefined instruction set (correction classes), which is highly language-dependent and involves the single task of restoring diacritics. On the other hand, our sequence-to-sequence ByT5 approach allows us to address multiple grammatical errors and learn to generate output sequences in a much more universal, language-independent approach.

### 4.3. Training Hyperparameters

The artificial neural networks are trained by updating their weights according to their response to the input. In particular, we focused on mini-batch gradient descent. For every mini-batch of *n* training examples (input $x^i$ and output $y^i$ pairs), the model parameters $\theta$ are updated using an objective function *J*:

$$\theta = \theta - \eta \cdot \nabla_\theta J(\theta; x^{i:i+n}; y^{i:i+n}). \tag{2}$$

The Adam [103] and Adafactor [104] extensions of this vanilla gradient descent are currently the most prevalent optimization algorithms for the transformer models. The success of training the models depends a lot on setting the hyperparameters in (2) correctly, such as the batch size *n*, the sequence length within a sample, and the learning rate $\eta$. We will discuss them in more detail.

### 4.3.1. Batch Size

This is the number of samples to be run through the model before updating the weights. The more tokens it has, the less disturbance an individual sample will cause during a (much smoother) weight update. On the other hand, very large batches take more time to compute and have diminishing gains.

The first popular pre-trained transformer, the BERT [89] model, for its classification, used a batch size of 256 sequences. A later model, RoBERTa [105], showed that an increase

in the batch size (up to 8000) and the dataset size accordingly improved the downstream performance. However, the same authors had to fine-tune the downstream applications using only batches of a size up to 48.

The popular seq2seq transformer, T5 [100], used batch size 128 for both pre-training and fine-tuning. Follow-up models, such as the multilingual version mT5 [101], the grammatical error correction model gT5 [106], and ByT5 [8] (the model we use in this work) all carried on with the same value for fine-tuning. The same size is also used in works solving the diacritics restoration task [47,107].

In conclusion, we can use a batch size of 128 or greater. All methods of this family use the same size and we are not strictly limited by the dataset size to increase it for better performance.

### 4.3.2. Maximum Sequence Length

When choosing the right batch size, one should also account for the maximum number of tokens allowed in a sample. There are two caveats here. First, the time complexity of the transformer model is quadratic on the sequence length $n$ (number of tokens) $\mathcal{O}(n^2)$, thus, shorter sequences are preferred for a faster training time. Secondly, the model we use operates in byte granularity and needs more tokens to express the same text, compared to word-level granularity models. The authors of the ByT5 model [8] report that English language sequences in byte tokens are about five times longer than in subword ones. As a result, the maximum sequence length for the ByT5 model is set to 1024 tokens. In our case, samples are sentences and, in practice, they all fit into this length.

### 4.3.3. Learning Rate

The last important parameter in (2) is the learning rate $\eta$. It controls how much the model parameters have to be updated. Low values of $\eta$ ensure smooth monotonic but small updates of the learned weights and a prolonged convergence. On the other hand, the higher learning rates would enlarge improvements and speed up the training. However, due to the higher "energy" (or "temperature") in the optimization, the high $\eta$ causes the "bouncing" of the learned parameter values and prevents settling in the best spot, resulting in the higher final training loss. An optimal learning rate value, as used during fine-tuning of the T5 family of models [8,100,101,106] with the Adafactor optimizer, is 0.001.

Sometimes, better results can be achieved by scheduling learning rate values during the training. There is, typically, the so-called warm-up period in the beginning to level discrepancies between previous parameters and new domain updates. It contains low or linearly increasing values of the learning rate. Similarly, as the training is to be finished, the "energy" of the optimization can be lowered by lowering the learning rate and allowing the neural network weights to settle in a more favorable position. As an example, during the original T5 [100] pre-training, a constant warm-up following an inverse square root decay with a peak learning rate of 0.01 was used. However, fine-tuning was performed with a constant value of 0.001. Such a learning rate is not dependent on the dataset size and it enables the straightforward comparisons of different setups. Overall, learning rate schedules can improve constant learning rate results, but they are less flexible to experiment with.

### 4.4. Evaluation

To evaluate diacritics restoration capabilities, we use the alpha-word accuracy metric from [38]. Each text sample is segmented into words, and for each word, we check if it is an alpha-word (alphabetical word):

- All characters in the word are alphabetic, where the general Unicode category property is one of "Lm", "Lt", "Lu", "Ll", or "Lo";
- It has at least one letter.

Given the number of gold (correct text) words to satisfy this condition $Tg$, as well as the number of these words that are correctly predicted by the system $Ts$, the alpha-word accuracy is

$$\text{alpha-word accuracy} = \frac{Ts}{Tg} \cdot 100\%. \tag{3}$$

This metric ensures that our results are not polluted by words that cannot have accents (e.g., numbers). Moreover, it takes into account both occasions of necessary and unnecessary accent generations. Other metrics, such as the Word Error Rate (WER) or the Diacritic Error Rate (DER), restrict themselves to $Tg$ of only the diacritized letters in the gold standard text [37].

## 5. Dataset

The expansion of the internet brought many abundant multilingual text resources. They usually vary from noisy and colossal to small in quantity but high in quality. A good example of the former is the Common Crawl dataset of more than 20TB of data, and its version OSCAR [108], which is filtered by language. Such huge datasets are now one of the main building blocks of the popular transformer models' pre-training, but they are very costly to work with during fine-tuning scenarios, such as our. The other extreme, such as the small high-quality Universal Dependencies [109] dataset, is too small to cover most aspects in each language.

Recent works on diacritics restoration seek a compromise between these two extremes. The authors of [48] use an OpenSubtitles dataset, which is of a satisfactory quality. On the other hand, the authors of [46] combine low-quality and high-quality datasets. They train first with the noisy web data, and finish with the higher quality Wikipedia dataset. However, training took two weeks for each language to reach the state-of-the-art results.

We use the same 12-language (Croatian, Czech, French, Hungarian, Irish, Latvian, Polish, Romanian, Slovak, Spanish, Turkish, and Vietnamese) Wikipedia dataset, proposed in [38]. Recent state-of-the-art diacritics restoration results were reported [46] for this dataset, so it is straightforward to compare with our methods on this particular task. As our focus is on efficiency, we omitted the large web text part to work only with the better-quality Wikipedia part.

We also add the Lithuanian language to the list, using the tools publicly provided by the original authors of [38] (we provide the links in the Data Availability Statement at the end of this article). The Lithuanian language is an omission we do not want to make here, not only because it is our mother tongue and, thus, we can interpret the results well, but also because it has some very unique features discussed in Section 5.1.

The dataset consists of training, development, and testing sets. All three are lower-cased, tokenized to words, and are split into sentences. The split between sets is performed on the Wikipedia article level. We show statistics of the training set in Table 1. The testing sets do not differ much, except that each language has exactly 30 000 sentences allocated to it and, thus, has a similar amount of words. The percentages of alpha-words, diacritic words, and diacritic letters in the testing sets do not deviate by more than 10%, compared to their training counterparts.

The dataset is already preprocessed to be used by simpler approaches, such as dictionary mapping. The ByT5 tokenization does not require that, as any text can be encoded in UTF-8 bytes; thus, it can work with any processed or unprocessed text.

**Table 1.** Languages and the training dataset statistics. Diacritic percentages are calculated among alphabetical words or letters. Alphabetical words (alpha-words) range from 72% to 86% of all the total words, including numbers.

| | Language | | | Dataset | | |
|---|---|---|---|---|---|---|
| **Name** | **Diacritic Letters** | **Keyboard Family** | **Sentences** | **Alpha-Words** | **Diacritic %** | |
| | | | | | **Words** | **Letters** |
| Croatian | 5 | QWERTZ | 802,610 | 12,914,186 | 14.55 | 2.78 |
| Czech | 19 | QWERTY | 952,909 | 14,730,260 | 48.69 | 12.90 |
| French | 15 | AZERTY | 1,818,618 | 37,612,736 | 16.49 | 3.72 |
| Hungarian | 9 | QWERTZ | 1,294,605 | 17,587,448 | 50.05 | 11.48 |
| Irish | 5 | QWERTY | 50,825 | 1,005,620 | 29.52 | 7.04 |
| Latvian | 15 | QWERTY | 315,807 | 4,244,914 | 48.57 | 10.27 |
| Lithuanian | 9 | QWERTY | 612,724 | 7,096,677 | 38.75 | 7.00 |
| Polish | 9 | QWERTY | 1,069,841 | 16,178,130 | 32.71 | 6.42 |
| Romanian | 6 | QWERTY | 837,647 | 16,050,136 | 27.04 | 5.87 |
| Slovak | 25 | QWERTZ | 613,727 | 9,180,800 | 42.38 | 9.32 |
| Spanish | 7 | QWERTY | 1,735,516 | 42,863,263 | 11.50 | 2.33 |
| Turkish | 11 | QWERTY | 875,781 | 10,785,516 | 31.35 | 6.30 |
| Vietnamese | 67 | QWERTY | 819,918 | 20,241,117 | 81.18 | 25.94 |

## 5.1. Features of Lithuanian

Here are some features of the Lithuanian language that make it interesting and important to include.

The Lithuanian language is highly inflective (fusional) and derivationally complex. It is different from agglutinative languages, that rely on prefixes, suffixes, and infixes. For inflections, Lithuanian "fuses" inflectional categories together, whereas prefixes, suffixes, and infixes are still used to derive words. For example, a diminutive/hypocoristic word can be derived by adding suffixes to the root, and the word can have two-three suffixes (sometimes going up to six), where each added suffix changes its meaning slightly. The language has compounds (connecting two-three words). Moreover, verbs can be made from any onomatopoeia; phrasal verbs (e.g., go in, go out) are composed by adding the prefix to the verb.

Some sentence structures are preferable in the Lithuanian language, but, syntactically, there is a lot of freedom in composing sentences. However, it is important to notice that the word order changes the sentence shade and message emphasis.

This complexity and variety of the forms makes isolated Lithuanian words ambiguous: 47% of Lithuanian word forms are morphologically ambiguous [110]. This, in turn, makes diacritic restoration and typo correction even more challenging.

## 5.2. A Realistic Model of Typos

We produce our pairs of correct (target) and incorrect (input) texts by taking the dataset as the correct (gold) text and by generating the corresponding incorrect text automatically.

The diacritic removal is straightforward, and is simply done by replacing all diacritic letters with the non-diacritic equivalents.

However, for typographical error inductions, a dedicated realistic corruption model is required. The approach, taken by other works [78,87], is to infer probabilities for each error group from the available smaller dataset and to use them to generate errors on the target one. We took the same approach in this work.

There are four prevailing categories of typographical errors. The authors of [60,111] reported that more than 80% of errors can be attributed to substitution, deletion, insertion, or transposition errors. This division allows us to model each category separately.

The physical keyboard layout plays an important role in influencing typos. A single keypress instruction consists of information of which hand, finger, and key row to select.

The authors of [53] argue that the confusion of these instructions is the main culprit of substitution errors, while mixed instruction timing between the two hands (operating on different parts of a keyboard) is the main culprit of transposition errors. While there may be more causes, such as visual and phonological factors [112], we restrict ourselves to the physical keyboard layout influence. This allows us to model typographical errors for all languages, given the distribution of the keyboard errors for a single language. We also make no distinction between physical and touchscreen keyboards, large or small.

There are only limited misspelling resources for the data-rich English language, as shown in Table 2. The largest one is the Github Typo corpus [113]. Although it contains edits for multiple languages, only the English language is of a significant size. There is also a multilingual Wikipedia edit history, which could be prepared, similar to the GitHub dataset. However, it must be filtered [114] to not include non-typographical error-related examples. Incorporating the Twitter Typo corpus [115] may also not be worth the effort, as the domains are different, as well as the length of text spans (needed to normalize error frequencies). In the end, we used a single GitHub Typo Corpus to derive the probabilities of errors.

**Table 2.** Related datasets for English misspelling corrections.

| Dataset | Number of Edits | Collection Method |
| --- | --- | --- |
| GitHub Typo Corpus [113] | 350,000 | Keyboard |
| Twitter Typo Corpus [115] | 39,171 | Keyboard |
| Birkbeck Spelling Corpus [116] | 36,133 | Handwritten |
| Holbrook Misspelling Corpus [117,118] | 1,791 | Handwritten |

Further details on generating the typos are provided in Section 6.2.

## 6. Experiment Details

Here, we provide further details on our experiments.

### 6.1. ByT5 Model Fine-Tuning

We chose the batch size of 256 and the default ByT5 maximum sequence length of 1024. Such a configuration matches the total maximum number of tokens ($256 \times 1024 = 2048 \times 128$) with the best system for diacritics restoration [46]. The larger sequence length is essential, as our model works on byte-level fine-grained tokens, compared to coarser subword-level models.

We used a GeForce RTX 2080 Ti GPU. Due to the modest memory size, we employed the gradient accumulation technique. It accumulates gradients in a continuous, rather than in a parallel, fashion. In addition, feeding only a single sample at a time allowed us to avoid padding.

We trained each model for 2048 steps, each consisting of 256 sentences/samples, with a total of $2048 \times 256 = 524{,}288$ sentences, and this took up to 10 h for a single model. For example, for the Lithuanian language, this corresponds to a 0.86 epoch over the total 612,724 sentences in its dataset (Table 1). In our results, we refer to such basic training as being trained for ×1 the number of sentences (#samples). We fix this training length, irrespective of the available dataset, for each language (Table 1) to make training comparable among languages. In experiments where we trained our models for longer (e.g., ×8), we used the whole dataset and passed through it as many times as needed, e.g., for Lithuanian ×8 corresponds to 6.8 epochs.

We used the Adafactor [104] optimizer with a constant learning rate of 0.001. The same setup was employed by the ByT5 [8] authors for fine-tuning experiments. Moreover, the Adafactor optimizer also has very little auxiliary storage compared to the other popular optimizer, Adam [103]. More complex learning rate schedules may give a slightly better

performance, but it would be more difficult to compare our runs, so we adhered to the constant learning rate approach.

For the diacritics restoration task with each language, we trained three different models. Each model has a different weight initialization, and data sampling is performed differently, according to a given random seed. The results are reported as a mean and a standard deviation over these three runs. In addition, we trained models for simultaneous diacritics and typographical error corrections for each language.

We also trained several models for a much longer time. First, we continued our basic fine-tuning setup with a batch size of 258 to 6000 steps (all other basic setups are up to 2048). At this stage, the loss became noisy (although it was low), so we increased our batch size to 8192 and continued training further. Due to the change in batch size, we reported our model training steps by how much training data, compared to our basic setup, it consumed. In our results, we reported models trained for ×8 and ×19 the number of samples in the basic setup. We chose those ceiling-rounded numbers as a means of convenience in our setup. As long training is very time-consuming, we performed only a few of them. We think that it still sufficiently indicates the scaling effects.

For text generation, in all our experiments we used a beam size of two. Later runs revealed that there is hardly a difference in size. As a result, for future work, we recommend adhering to a simpler beam size of 1.

The training script and the Pytorch model implementation were used from the Hugging Face library [95]. If not stated otherwise, we used all default parameters as they are in this library version 4.12.0.

*6.2. The Generation of Typographical Errors*

We took a similar approach for the generation of typographical errors, as in [78]. Close to a process of text writing, the program moves through each symbol and induces errors in a stochastic manner by evaluating probabilities of various error types for each character. This includes deletion, insertion, substitution, and transposition operations.

The chance for a letter to participate in a particular error type is determined according to the frequency of errors in the reference dataset. We used the largest known original typo dataset, the GitHub Typo Corpus [113]. The dataset was filtered for only English language typos and the characters were selected with a count of at least 1000. Given the final character set $C$, the total number of times $f(c)$ the character $c \in C$ or a specific typo pattern appeared in the selected corpus, the following probabilities for each character are considered:

$$P(\text{deletion} \mid c) = \frac{f(c \to \varnothing)}{f(c)}, \tag{4}$$

$$P(\text{substitution} \mid c) = \frac{\sum\limits_{\underline{c} \in C} f(c \to \underline{c})}{f(c)}, \tag{5}$$

$$P(\text{insertion after} \mid c) = \frac{\sum\limits_{\underline{c} \in C} f(c \to c\underline{c})}{2f(c)}, \quad P(\text{insertion before} \mid c) = \frac{\sum\limits_{\underline{c} \in C} f(c \to \underline{c}c)}{2f(c)}, \tag{6}$$

$$P(\text{transposition} \mid cc') = \frac{f(cc' \to c'c)}{f(cc')}. \tag{7}$$

Note that we divide insertion errors into two distinct categories, whether the character is inserted after the one in question, or before. Both insertion probabilities are collected from the same samples, so we divide them by two. An alternative way would be to collect triplets of characters before the one in question and after, but the probabilities would then be sparse. Nevertheless, our chosen approach covers the so-called "fat-finger" errors.

We ran some typographical error induction experiments on the original GitHub Corpus and confirmed that our generation method aligns with the original error type distribution. Initially, only about 1% of characters were corrupted, so we scaled our probabilities by

a factor of three to be close to the low error rate, as defined in [78]. The final error type distribution and percentage of the corrupted characters for each language are depicted in Figure 1. The amount of generated errors for each language slightly varies because the letter frequencies derived from English differ in other languages.

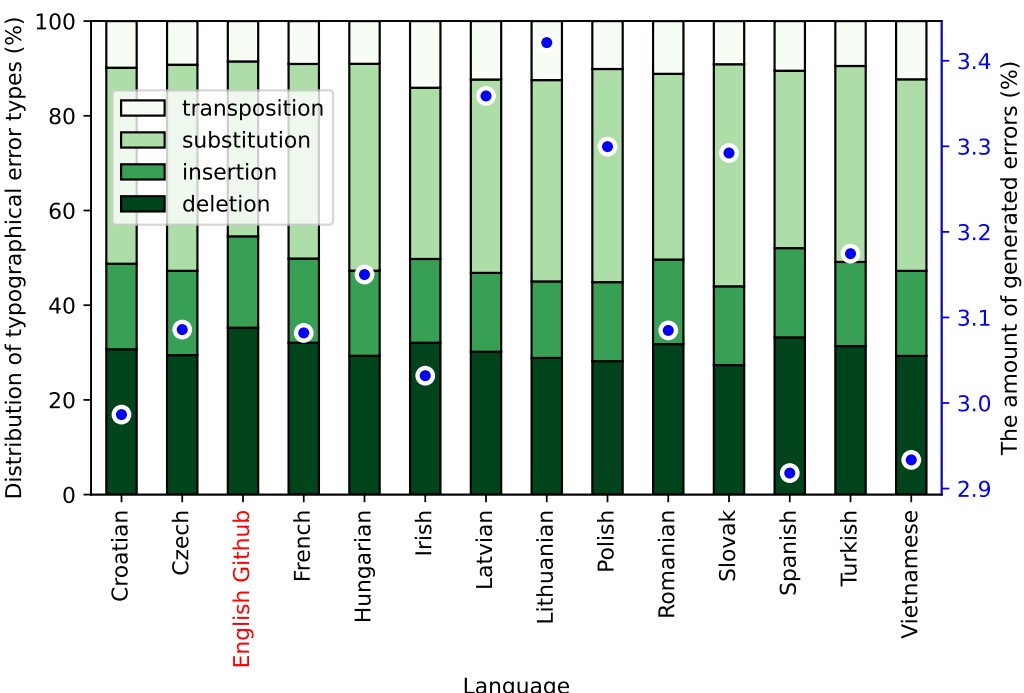

**Figure 1.** Distribution of generated typographical errors by category (the left vertical axis and stacked bars). Proportions for the English part of the GitHub Corpus (used to derive generation probabilities) are also depicted for reference. The total percentage of induced corruptions are included (the right vertical axis and corresponding blue dots).

Insertion and substitution errors can result in many different outcomes. The probabilities for specific letters to emerge, given that this type of error occurs at a specific place, are computed by the following equations:

$$P(c \rightarrow c' \mid c, \text{substitution}) = \frac{f(c \rightarrow c')}{\sum\limits_{\underline{c} \in C} f(c \rightarrow \underline{c})}, \tag{8}$$

$$P(c \rightarrow cc' \mid c, \text{insertion after}) = \frac{f(c \rightarrow cc')}{\sum\limits_{\underline{c} \in C} f(c \rightarrow c\underline{c})}, \tag{9}$$

$$P(c \rightarrow c'c \mid c, \text{insertion before}) = \frac{f(c \rightarrow c'c)}{\sum\limits_{\underline{c} \in C} f(c \rightarrow \underline{c}c)}. \tag{10}$$

As mentioned previously, we took the typo statistics from the English dataset and ran on the assumption that typos are based purely on the layout of the keyboard (the proximity of keys, etc.), so the same typo statistics will be in all the other languages using the QWERTY layout. We did not deal with the extensions of the character sets and keyboard layouts for different languages, as we only introduced typos to the undiacritized versions of the texts, irrespective of the case. We disregarded other possible minor variations in the keyboard layouts as insignificant.

For the Croatian, French, Hungarian, and Slovak languages, corresponding to their different keyboard layout families (see Table 1), we remapped the original English QWERTY

dataset before inferring typo probabilities. For example, for Croatian, which has a QWERTZ layout, we had to swap the letters "z" and "y" when calculating probabilities. In our initial experiments, we did not observe significant model performance differences between the QWERTY and remapped typo generation versions.

## 7. Results

We present the results of our different experiments here.

### 7.1. Diacritics Restoration

The diacritics restoration results are presented in Table 3. Our ByT5 method results lay between the dictionary (a simple statistical Unigram model) and the state-of-the-art model [46]. The highest alpha-word accuracy is for French, Spanish, and Croatian, with results that were only 0.34%, 0.29%, and 0.56% behind the state of the art, respectively. These languages have the smallest percentage of diacritic words (see Table 1). The lowest scores are recorded for Vietnamese and Latvian at 94.25% and 96.33%, respectively. We also note that the Irish language, with the smallest dataset, has the highest standard deviation of 0.32%.

The "Raw" column in Table 3 indicates the alpha-word accuracy of the uncorrected text for comparison. Naturally, the more diacritic-heavy the language is, the lower the number.

An Approach with the Dictionary and the ByT5 models (Dict.+ByT5)

We noticed that the dictionary method outperforms the ByT5 method for words that have only a single target translation in the dictionary. We grouped words by how many translation targets in the dictionary they have and we show the ratio of ByT5-to-Dictionary error rates in Table 4. The resulting values that are higher than 1 indicate the Dictionary outperforming the ByT5 model. This is the case for all languages at a word group with only a single translation.

**Table 3.** Alpha-word accuracy results (%) for the diacritics restoration task. We report means and standard deviations for three separate training runs with different initial model weights and dataset samplings trained for 524 288 sentences (#samples: ×1) and a single run for eight times more(×8), cycling through the available training data (Table 1) as needed.

| Language | Raw | Dict. | [46] | ByT5 #samples: ×1 | ×8 | Dict.+ByT5 ×1 | ×8 |
|---|---|---|---|---|---|---|---|
| Croatian | 85.01 | 99.11 | 99.73 | $99.17 \pm 0.06$ | | $99.42 \pm 0.03$ | |
| Czech | 49.71 | 95.67 | 99.22 | $98.01 \pm 0.03$ | | $98.38 \pm 0.04$ | |
| French | 83.11 | 97.98 | 99.71 | $99.37 \pm 0.04$ | | $99.49 \pm 0.03$ | |
| Hungarian | 50.34 | 96.22 | 99.41 | $98.42 \pm 0.02$ | 99.20 | $98.78 \pm 0.01$ | 99.25 |
| Irish | 69.97 | 96.65 | 98.88 | $98.14 \pm 0.32$ | | $98.40 \pm 0.16$ | |
| Latvian | 50.14 | 90.59 | 98.63 | $96.33 \pm 0.12$ | 97.78 | $96.62 \pm 0.09$ | 97.66 |
| Lithuanian | 60.76 | 93.83 | — | $97.94 \pm 0.19$ | 99.07 | $98.18 \pm 0.13$ | 98.95 |
| Polish | 66.73 | 97.00 | 99.66 | $99.00 \pm 0.03$ | | $99.16 \pm 0.02$ | |
| Romanian | 70.37 | 96.09 | 98.64 | $97.99 \pm 0.03$ | | $98.17 \pm 0.04$ | |
| Slovak | 56.34 | 96.88 | 99.32 | $98.43 \pm 0.06$ | | $98.77 \pm 0.02$ | |
| Spanish | 87.97 | 99.11 | 99.62 | $99.33 \pm 0.04$ | | $99.43 \pm 0.02$ | |
| Turkish | 68.39 | 98.41 | 98.95 | $98.86 \pm 0.04$ | | $99.03 \pm 0.02$ | |
| Vietnamese | 15.88 | 73.53 | 98.53 | $94.25 \pm 0.07$ | 97.53 | $94.29 \pm 0.07$ | 97.54 |
| Average | 62.67 | 94.70 | 99.19 | 98.10 | | 98.32 | |

**Table 4.** Alpha-word error ratio between the ByT5 and Dictionary methods for two word groups and models in different training stages. The values higher than 1 indicate that the Dictionary method restores diacritics better. The first word group corresponds to words with exactly one possible translation target, and the second word group corresponds to words with two translation targets. Groups are determined by the training set statistics, while results are reported on the testing set.

| Language | One Dictionary Candidate | | | Two Dictionary Candidates | | |
|---|---|---|---|---|---|---|
| #samples: | ×0.5 | ×1 | ×8 | ×0.5 | ×1 | ×8 |
| Croatian | $6.37 \pm 0.98$ | $4.98 \pm 0.52$ | | $1.18 \pm 0.03$ | $1.01 \pm 0.06$ | |
| Czech | $4.74 \pm 0.19$ | $3.53 \pm 0.03$ | | $0.45 \pm 0.02$ | $0.37 \pm 0.01$ | |
| French | $5.29 \pm 0.17$ | $4.98 \pm 0.48$ | | $0.31 \pm 0.02$ | $0.27 \pm 0.01$ | |
| Hungarian | $7.35 \pm 0.48$ | $4.37 \pm 0.10$ | 1.42 | $0.84 \pm 0.03$ | $0.62 \pm 0.00$ | 0.34 |
| Irish | $2.13 \pm 0.21$ | $2.27 \pm 0.84$ | | $0.56 \pm 0.03$ | $0.57 \pm 0.08$ | |
| Latvian | $2.43 \pm 0.17$ | $1.77 \pm 0.11$ | 0.69 | $0.43 \pm 0.00$ | $0.37 \pm 0.02$ | 0.23 |
| Lithuanian | $2.61 \pm 0.18$ | $2.00 \pm 0.36$ | 0.58 | $0.34 \pm 0.02$ | $0.27 \pm 0.01$ | 0.12 |
| Polish | $4.06 \pm 0.24$ | $2.56 \pm 0.15$ | | $0.30 \pm 0.03$ | $0.24 \pm 0.01$ | |
| Romanian | $3.66 \pm 0.44$ | $2.63 \pm 0.12$ | | $0.82 \pm 0.02$ | $0.63 \pm 0.02$ | |
| Slovak | $4.29 \pm 0.03$ | $3.00 \pm 0.21$ | | $0.54 \pm 0.02$ | $0.44 \pm 0.01$ | |
| Spanish | $5.4 \pm 0.54$ | $4.18 \pm 0.57$ | | $0.95 \pm 0.04$ | $0.82 \pm 0.03$ | |
| Turkish | $10.5 \pm 10.84$ | $2.70 \pm 0.24$ | | $3.83 \pm 4.68$ | $1.01 \pm 0.03$ | |
| Vietnamese | $2.6 \pm 0.10$ | $2.38 \pm 0.20$ | 1.31 | $1.52 \pm 0.16$ | $1.21 \pm 0.04$ | 0.32 |

Table 4 also portrays how the ratio of the ByT5-to-Dictionary error rates changes during half and full training. The trend is obvious: the transformer improves for all word groups with training. If our training was longer, the ByT5 model may even surpass the Dictionary model at a word group of one translation candidate. This is exactly what happened for the Latvian and the Lithuanian languages after eight times more training samples.

Note that at half the training, the standard deviation of the Turkish ratio is abnormally high. This is due to one of three ByT5 training runs that temporarily fail. However, with further training, the run recovered up to the same accuracy level as the other two. This is a good example of how different training dynamics can be dependent on different initial conditions and different data sampling.

We constructed a hybrid approach by letting the Dictionary model restore words with only a single translation candidate, while leaving all the other words for the transformer. For our standard training, this improved the single ByT5 results by up to 0.37%, on average, and allowed us to reach the state-of-the-art results for the Turkish language. However, we can observe that, with longer training, the pure ByT5 model can catch up to, or even surpass, the hybrid approach.

### 7.2. Simultaneous Diacritics and Typos Corrections

The results of the simultaneous diacritic and typographic error corrections are represented in Table 5. We see that the alpha-word accuracy results are significantly lower across the board, compared to restoring the diacritics alone.

**Table 5.** Alpha-word accuracies for the simultaneous diacritics and typographic error corrections.

| Language | Raw | Hunspell | Dict. | ByT5 | | Dict.+ByT5 | |
|---|---|---|---|---|---|---|---|
| | | | | #samples: ×1 | ×19 | ×1 | ×19 |
| Croatian | 64.05 | 66.43 | 74.06 | 90.27 | | 96.71 | |
| Czech | 38.68 | 40.15 | 71.37 | 89.88 | | 94.52 | |
| French | 60.81 | 64.94 | 70.87 | 93.45 | | 96.52 | |
| Hungarian | 38.16 | 46.35 | 69.84 | 88.31 | 93.96 | 94.31 | 96.85 |
| Irish | 53.49 | 56.01 | 73.16 | 89.48 | | 94.49 | |
| Latvian | 37.69 | 44.21 | 66.29 | 88.88 | | 93.01 | |
| Lithuanian | 44.78 | 44.87 | 68.44 | 89.68 | 94.19 | 94.70 | 96.73 |
| Polish | 49.10 | 56.61 | 70.02 | 91.38 | | 96.76 | |
| Romanian | 51.93 | 54.54 | 70.29 | 90.50 | | 94.14 | |
| Slovak | 43.92 | 48.59 | 72.89 | 91.05 | | 95.56 | |
| Spanish | 64.07 | 68.03 | 71.58 | 93.12 | | 95.98 | |
| Turkish | 51.18 | 51.69 | 72.58 | 90.00 | | 95.29 | |
| Vietnamese | 11.92 | 11.84 | 56.19 | 87.34 | 93.40 | 87.86 | 93.77 |
| Average | 46.91 | 50.33 | 71.89 | 90.26 | | 94.60 | |

We also added correction results that were obtained with the open-source Hunspell spellchecker [52] by replacing the words that it found to be incorrectly spelled with its first suggestion. The results indicate that it is barely better than raw uncorrected sentences. It is also significantly worse than our Dictionary approach, which is specialized in restoring diacritics.

The Dictionary method was used in the same way as the previous experiment, i.e., it was "trained" on the typo-free diacritization-only task in both the standalone and hybrid approaches.

The reduction of accuracy the ByT5 model, on average, is by 7.84%, while for hybrid Dict.+ByT5 approach, it was 3.71%. A smaller reduction for the hybrid method suggests that the transformers do not cope well with the same words that it successfully dealt with when there were no typos present. A possible reason may be that more learning is required by both tasks, and up to 10 h of training might not be enough. Training the Hungarian model up to 19 times longer improves the performance substantially, but the gap of 2.98% between the ByT5 model and the hybrid remains.

### 7.3. Performance on the Zipf's Tail

Word frequencies can be modeled reasonably well by a Zipf distribution. It is a very heavy-tailed distribution, where there is a vast number of words with low frequencies. The abundance of such words is a challenge for most learning systems, as the data for these points is sparse. Our question is, how hard are these words for our trained models?

We grouped words that were in our testing set by their frequencies in the training set. The resulting word groups are:

- Unseen: present in the test but not in train data;
- [1, 100]: words appearing in the training set from 1 to 100 times;
- [101, 10,000]: words appearing in training set from 101 to 10,000 times.

Alpha-word accuracy results for these groups are shown in Table 6.

**Table 6.** The distribution of diacritics restoration errors for the different frequencies of the words in the training dataset are shown in the first three columns. The intervals indicate the bounds of how many times the words in this group were encountered in the training dataset. The last two columns indicate the percentages of how much of the training dataset was constituted by these word groups.

| Language | Percentage of All Errors | | | % of Training Dataset | |
|---|---|---|---|---|---|
| | Unseen | [1, 100] | [101, 10,000] | [1, 100] | [101, 10,000] |
| Croatian | 31.42 ± 1.65 | 54.23 ± 1.27 | 12.67 ± 1.45 | 11.79 | 51.02 |
| Czech | 20.42 ± 0.24 | 51.07 ± 1.31 | 26.43 ± 0.80 | 10.62 | 54.54 |
| French | 13.55 ± 0.88 | 42.29 ± 3.26 | 28.53 ± 1.66 | 2.06 | 37.53 |
| Hungarian | 26.78 ± 0.33 | 49.96 ± 0.59 | 19.67 ± 0.69 | 9.25 | 52.82 |
| ×8 #samples | 35.97 | 43.45 | 17.64 | | |
| Irish | 46.51 ± 6.04 | 35.37 ± 0.65 | 13.64 ± 3.17 | 23.53 | 45.03 |
| Latvian | 25.94 ± 0.95 | 49.87 ± 0.16 | 21.35 ± 0.77 | 22.20 | 56.23 |
| ×8 #samples | 33.71 | 43.23 | 20.49 | | |
| Lithuanian | 26.13 ± 0.60 | 47.93 ± 1.03 | 24.83 ± 1.11 | 16.41 | 63.61 |
| ×8 #samples | 40.28 | 39.60 | 19.55 | | |
| Polish | 18.31 ± 0.52 | 48.49 ± 0.89 | 29.78 ± 0.46 | 10.43 | 56.29 |
| Romanian | 16.06 ± 0.29 | 43.46 ± 0.96 | 27.83 ± 1.08 | 6.83 | 48.39 |
| Slovak | 36.68 ± 1.00 | 52.96 ± 0.29 | 10.01 ± 0.70 | 14.98 | 52.18 |
| Spanish | 13.22 ± 0.24 | 39.29 ± 1.58 | 28.74 ± 0.86 | 2.19 | 36.81 |
| Turkish | 24.11 ± 0.48 | 54.03 ± 1.13 | 21.07 ± 0.71 | 11.26 | 59.66 |
| Vietnamese | 1.48 ± 0.01 | 7.85 ± 0.19 | 58.40 ± 0.46 | 0.96 | 26.15 |
| ×8 #samples | 3.37 | 10.69 | 55.43 | | |

A substantial part of errors come from the words that are unseen during the training. Excluding Vietnamese and Irish, this ranges from 13% (Spanish, French) to 36% for Slovak. The Vietnamese outlier of 1% may be due to its linguistic nature, while the Irish outlier of 46% is due to its very small dataset. Overall, the smaller the dataset (Table 1), the more unseen or rare words, and the associated errors, we have.

Similar to the Dictionary method and the other classical methods, unseen data is also a significant source of errors for the transformer model. Different to the classical approaches, however, is the transformer model, which is based on neural networks, and it can generalize to unseen data. To investigate this generalization, we filtered all the words that were in the testing set and not in the training and calculated the percentages, as is shown in Table 7. We can see that the ByT5 model successfully restores more than 76% of unseen words for each language.

**Table 7.** The confusion matrix of the unseen word diacritics restoration performance by the ByT5 model. Unseen words with and without diacritics are presented separately. The last column depicts the total number of unseen words for each language.

| Language | With Diacritics, % | | Without Diacritics, % | | Total Unseen |
| --- | --- | --- | --- | --- | --- |
| | **Failed** | **Restored** | **Failed** | **Left Correct** | |
| Croatian | $6.8 \pm 0.2$ | $15.9 \pm 0.2$ | $4.3 \pm 0.4$ | $73.0 \pm 0.4$ | 12,147 |
| Czech | $16.4 \pm 0.2$ | $37.7 \pm 0.2$ | $5.0 \pm 0.4$ | $40.9 \pm 0.4$ | 9398 |
| French | $10.1 \pm 0.1$ | $9.8 \pm 0.1$ | $4.6 \pm 0.3$ | $75.6 \pm 0.3$ | 3794 |
| Hungarian | $10.1 \pm 0.2$ | $58.2 \pm 0.2$ | $2.3 \pm 0.1$ | $29.5 \pm 0.1$ | 16,350 |
| $\times 8$ #samples | 7.1 | 61.2 | 1.2 | 30.5 | |
| Irish | $12.1 \pm 0.5$ | $25.6 \pm 0.5$ | $5.3 \pm 1.0$ | $57.0 \pm 1.0$ | 29,470 |
| Latvian | $16.8 \pm 0.7$ | $39.9 \pm 0.7$ | $5.5 \pm 0.6$ | $37.9 \pm 0.6$ | 17,449 |
| $\times 8$ #samples | 13.6 | 43.1 | 3.9 | 39.4 | |
| Lithuanian | $8.1 \pm 0.4$ | $29.7 \pm 0.4$ | $4.4 \pm 1.4$ | $57.7 \pm 1.4$ | 15,547 |
| $\times 8$ #samples | 6.0 | 31.9 | 2.8 | 59.3 | |
| Polish | $7.0 \pm 0.1$ | $20.4 \pm 0.1$ | $2.3 \pm 0.2$ | $70.3 \pm 0.2$ | 9461 |
| Romanian | $15.6 \pm 0.5$ | $15.1 \pm 0.5$ | $6.9 \pm 0.9$ | $62.4 \pm 0.9$ | 8493 |
| Slovak | $14.4 \pm 0.3$ | $33.7 \pm 0.3$ | $4.9 \pm 1.6$ | $46.9 \pm 1.6$ | 13,357 |
| Spanish | $8.1 \pm 0.2$ | $11.6 \pm 0.2$ | $5.4 \pm 1.0$ | $75.0 \pm 1.0$ | 5115 |
| Turkish | $7.0 \pm 0.1$ | $25.3 \pm 0.1$ | $3.7 \pm 0.2$ | $63.9 \pm 0.2$ | 9594 |
| Vietnamese | $14.4 \pm 0.3$ | $1.7 \pm 0.3$ | $1.9 \pm 0.4$ | $82.1 \pm 0.4$ | 4260 |
| $\times 8$ #samples | 13.1 | 2.9 | 2.7 | 81.2 | |

### 7.4. Training Longer

Training for longer is beneficial. As can be seen in Figure 2, testing the alpha-word accuracy for all our models is only increaing with training. The lack of training hurts the performance of the Vietnamese language the most, which is the language with the most diacritics. Training the corresponding model for eight times longer brings substantial improvements of over 3.28%.

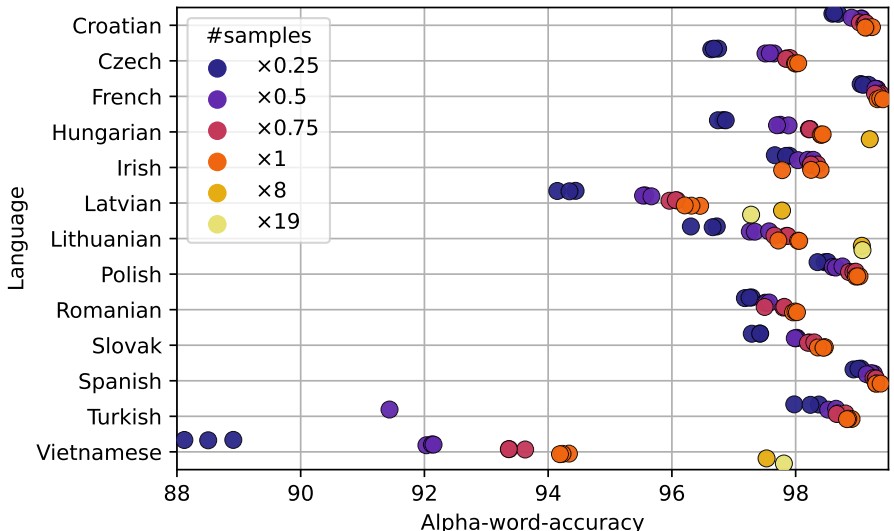

**Figure 2.** Alpha-word accuracy improvement during diacritics restoration training. Training steps of $\times 1$ corresponds to $2048 \times 256$ sentences for a given language. There is a visible outlier for the Turkish language at $\times 0.5$ training steps, but that model regained the accuracy later in training.

A similar trend is observed for all the models trained on the two tasks simultaneously in Figure 3. Here, the improvements are much larger. On the other hand, languages with fewer diacritics, such as French and Spanish, have diminishing gains from longer training. Overall, longer training is a must for the more difficult tasks.

Note that while the training is much longer, we still use the same dataset sizes presented in Table 1, but we just iterate over them more times.

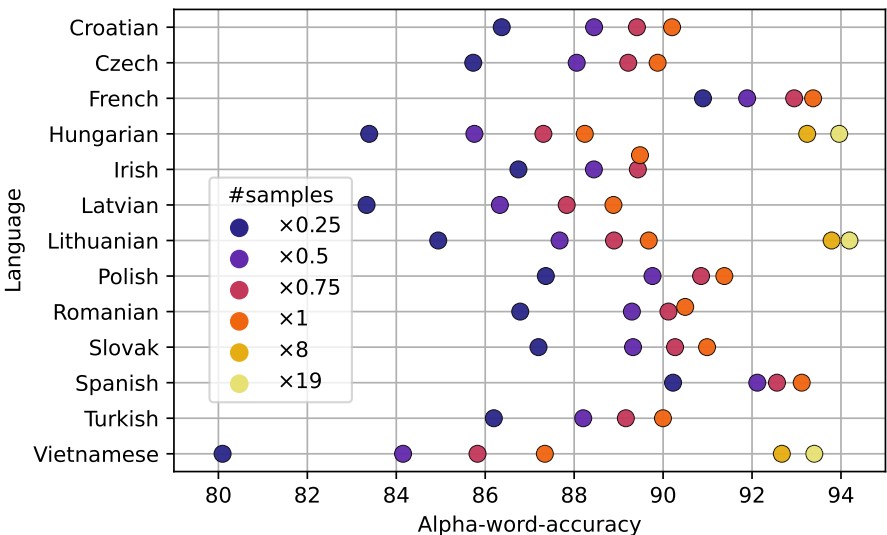

**Figure 3.** Alpha-word accuracy improvement during diacritics and typographical errors correction training. Training data of ×1 corresponds to 2048 × 256 sentences for a given language. We also run a single longer training session for the Hungarian language, with up to ×19 training steps.

## 8. Discussion

In this work, we show that accuracy can be improved by combining the transformer and the classical Dictionary methods. Yet, this is the case for more under-trained transformers. We show that the longer-trained ByT5 models start to bypass the hybrid approach. However, when resources are limited compared to the difficulty of the task, such a hybrid approach can be a viable solution, as is the case with our simultaneous diacritics restoration and typos correction tasks.

The hybrid Dict.+ByT5 approach might also have an advantage in the latter task because the dictionary part is "trained" on the typo-free diacritization task and, thus, recognizes and corrects typo-free words well. The ByT5 model was trained only on the combined task, so it, thus, has a harder time learning to recognize these situations from the noisy data.

Transformer models depend on the amount of training data, and small sizes can hinder the performance. Hungarian and Latvian languages, with a very similar percentage of diacritics (and, hence, the task difficulty), had a difference of four times between their dataset sizes. As a result, our achieved restoration score for Latvian was almost 2% lower. On the other hand, the alpha-word accuracy of over 96% and 98% can still be reached for Latvian and Irish languages, with dataset sizes of 5.5 M and 1.2 M words, respectively. This indicates a correlation between the difficulty of the task and the size of the dataset needed.

One way to improve our results is to leverage the fact that most of the errors are due to unseen and less-seen words in the training data. As we show in this work (Table 6), longer training improves the restoration of words with moderate frequencies but it is less effective for unseen words and is very time-consuming. The only way to improve unseen words is to rely on the additional dataset. Time constraints could, additionally, be relieved by employing boosting approaches [119], i.e., training on the filtered selection of data, which is known to be problematic. Such data could contain a high proportion of low-frequency and unseen words, while at the same time, being compact.

A limitation of our work is that we had only a single moderate GPU at our disposal. Scaling the model size [106], incorporating additional datasets [46], and training longer can improve accuracy by several percent. Similarly, one can build a model of multiple languages to gain benefits by overlapping vocabularies and semantics of related under-represented languages, although studies report contradictory results [46,48]. We think that all these scaling approaches are promising as future work.

In our work, we generated the typos for the entire datasets just once, but, in principle, we could generate different typos each time we pass through the dataset. This would require more computation, but it would enrich the data for longer training sessions.

Another natural future direction is the incorporation of multiple error types. This is still an active area of research, as the currently achievable accuracy of such systems has a wide margin to improve [107]. In this work, we show how difficult the task becomes by combining just two classes of errors. However, this is a bigger problem for the classical hand-crafted approaches, but our ByT5-based models could, in principle, cope with this, given additional data and training times.

Our approach is also easy to scale to other languages, as it does not depend on the alphabet or structure of the language. For example, only the typo dataset generation model in this work depends on the Latin alphabet and a corresponding keyboard layout.

Altogether, this makes our approach very promising for large-scale real-world applications. Our combined diacritic restoration and typo correction solution could, in principle, already be used in, for example, auto-correcting text messages or social media posts/comments. Expanding the approach in the ways discussed above opens even bigger application horizons.

## 9. Conclusions

We achieved a 98.3% average alpha-word accuracy (within 1% of the state of the art) on the diacritic restoration task over 13 benchmark languages with a ByT5 universal byte-level transformer model approach, a smaller training dataset (Wikipedia), and a much-reduced training time (Table 3). When the training time is limited, the model is slightly improved by the assistance of a simple statistical Unigram model (Dict.+ByT5). There is a solid indication, however, that longer training gets very close to the state-of-the-art model, even without this assistance, and with the smaller dataset (Figure 2).

We achieved a 94.6% average alpha-word accuracy on the simultaneous diacritics restoration and typo correction tasks with the same models (Dict.+ByT5), training datasets and times. This is a much harder task, and is problematic for the specialized systems; thus, we have no state-of-the-art model to compare to (Table 5). There is also a strong indication that longer training can significantly improve these results (Figure 3).

We investigated that most of the errors are caused by the words that are rare in the training dataset (Table 6). However, contrary to the classical approaches, our models generalize quite well to the unseen words (Table 7) and restore diacritics correctly on >76% of the unseen words in every language. This gives us good hints on how the models can be further improved, often by simply training them more.

The good performance and universality of this approach make it very promising for real-world applications, more languages and error classes.

**Author Contributions:** Conceptualization, M.L., J.K.-D., L.S. and T.K.; methodology, L.S., M.L., J.K.-D. and M.B.; software, L.S.; validation, L.S.; formal analysis, L.S. and M.L.; investigation, L.S.; resources, M.L.; data curation, L.S.; writing—original draft preparation, L.S., M.L., J.K.-D. and M.B.; writing—review and editing, M.L. and J.K.-D.; visualization, L.S.; supervision, M.L. and J.K.-D.; project administration, M.L. and J.K.-D.; funding acquisition, M.L., J.K.-D., L.S., T.K. and M.B. All authors have read and agreed to the published version of the manuscript.

**Funding:** This research was funded by the joint Kaunas University of Technology Research and Innovation Fund and Vytautas Magnus University project "Deep-Learning-Based Automatic Lithuanian Text Editor (Lituanistas)", Project No.: PP34/2108.

**Institutional Review Board Statement:** Not applicable.

**Informed Consent Statement:** Not applicable.

**Data Availability Statement:** Publicly available datasets were analyzed in this study. The data for 12 benchmark languages can be found here: http://hdl.handle.net/11234/1-2607. Additional data for the Lithuanian were used from here: https://ufal.mff.cuni.cz/~majlis/w2c/download.html and were preprocessed by the tools from https://github.com/arahusky/diacritics_restoration/tree/master/data/create_corpus_scripts. All the links were last accessed on 3 January 2022.

**Conflicts of Interest:** The authors declare no conflict of interest.

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
