# Peer review of "Correcting Diacritics and Typos with a ByT5 Transformer Model"

_applsci, doi:10.3390/app12052636_

Round 1

Reviewer 1 Report

This paper presents their work using ByT5 transformer model of tackling diacritics restoration and spelling correction simultaneously. Extensive experiments are done on datasets of 13 languages and 12 benchmark datasets with an additional dataset of the Lithuanian language with different settings, including a setting with fewer data. Near state-of-the-art performance is achieved in most settings. It's suggested that further improvement can be achieved with longer training. The work is thoroughly done and nicely described. It would be better if some error analysis is presented. 

Author Response

Dear reviewer,

Thank you for your time and the positive review. 

> This paper presents their work using ByT5 transformer model of tackling diacritics restoration and spelling correction simultaneously. Extensive experiments are done on datasets of 13 languages and 12 benchmark datasets with an additional dataset of the Lithuanian language with different settings, including a setting with fewer data. Near state-of-the-art performance is achieved in most settings. It's suggested that further improvement can be achieved with longer training. 
> The work is thoroughly done and nicely described. 

Thank you.

> It would be better if some error analysis is presented. 

Thank you for your suggestion. We analyzed diacritic restoration errors by the corresponding word frequencies in the training set (Table 6) and by whether words contain diacritics or not (Table 7), also commented on possible reasons in the text.

Sincerely,

Authors

Reviewer 2 Report

The authors proposed a manuscript on correcting diacritics and typos with ByT5 Transformer Model. However, there are some revisions that need to be addressed before the manuscript could be accepted.

  1. The authors should better emphasize the background behind their proposed model and provide more reasoning
  2. The authors should provide experiments that compare their proposed model with other previous works with similar models
  3. The authors should add an ablation study to the manuscript
  4. The proposed architecture should be drawn in more detail, including encoder and decoder model, accompanied by mathematical models
  5. Provide more reasoning on the language comparison, especially why Lithuanian is selected, is there any special characteristics compared with other languages
  6. Other keyboard layouts besides qwerty could also be considered and analyzed
  7. Provide more explanation on the transformer model ByT5, compared with Bert or Roberta
  8. Provide more explanation on data validation used in the experiments
  9. Add explanation on real-world application of this model

Author Response

Dear reviewer,

Thank you for your time and your help to improve the paper. Below we have responded to all of your remarks/comments. 

> The authors proposed a manuscript on correcting diacritics and typos with ByT5 Transformer Model. However, there are some revisions that need to be addressed before the manuscript could be accepted.

Thank you for the valuable suggestions.

> 1.  The authors should better emphasize the background behind their proposed model and provide more reasoning

We now explained the difference between our ByT5-based approach from the previous state-of-the-art BERT-based approach [46] better. It is added at the end of Section 4.2.4.:

The sequence-to-sequence nature of ByT5 tackles the limitations of the latest state-of-the-art diacritics restoration [46] which is based on BERT. The latter system was of an auto-encoding type and performed classification for each token. I.e., it had to predict the proper classes of each token correction, described by the position and diacritic sign type. This system is limited to its predefined instruction set (correction classes), which is highly language-dependent, and the single task of restoring diacritics. On the other hand, our sequence-to-sequence ByT5 approach allows us to address multiple grammatical errors and learn to generate output sequences in a much more universal, language-independent approach.

> 2.  The authors should provide experiments that compare their proposed model with other previous works with similar models

For a single diacritics restoration task, we compared our results to [46]. These comparisons were made in 12 languages, with the same testing set and with a comparable transformer architecture model.

For both diacritics and typos restoration tasks, although there are some published research efforts such as [106], we could not find any credible publicly available model weights. As a result, the comparison would require training these models from scratch which is prohibitively expensive. Moreover, articles such as [106], do not cover even half of our 12+1 languages. Eventually, we compared it with a classical dictionary method (see Table 5). 

To make these comparisons even more grounded, we now added results with a Hunspell spell checker (see Table 5).

> 3.  The authors should add an ablation study to the manuscript

We have investigated how the performance changes with the ByT5 in combination with dictionary methods, compared to the ByT5 model alone, also how well the models perform depending on the amount of training, testing with 13 different languages and two different tasks. We briefly discuss the effects on the performance of details such as batch size and beam size in Section 6.1.

We have also added a comparison with a classical spellchecker.

Unfortunately, the combinatorics of all the possible model, training, and experiment variations is too high and the experiments take too long to add more studies or meaningful systematic reports in this short time.

> 4.  The proposed architecture should be drawn in more detail, including the encoder and decoder model, accompanied by mathematical models

We did not include the details of the ByT5 model, because it is not created by us and is well-explained in the original referenced contributions. We added, however, a better explanation of why its use is significant and different from the previous state-of-the-art in our case at the end of Section 4.2.4.:

The sequence-to-sequence nature of ByT5 tackles the limitations of the latest state-of-the-art diacritics restoration [46] which is based on BERT. The latter system was of an auto-encoding type and performed classification for each token. I.e., it had to predict the proper classes of each token correction, described by the position and diacritic sign type. This system is limited to its predefined instruction set (correction classes), which is highly language-dependent and is the single task of restoring diacritics. On the other hand, our sequence-to-sequence ByT5 approach allows us to address multiple grammatical errors and learn to generate output sequences in a much more universal, language-independent approach.

> 5.  Provide more reasoning on the language comparison, especially why Lithuanian is selected, is there any special characteristics compared with other languages

We choose the same 12 languages that were used in the state-of-the-art diacritics restoration [46] to make our results directly comparable.

We now added motivation for including the Lithuanian language:

In Section 5:

We also add the Lithuanian language to the list, using the tools publicly provided by the original authors of [38] (we provide the links in the data availability statement at the end of this article). The Lithuanian language is an omission we do not want to make here, not only because it is our mother tongue and thus we can interpret the results well, but also because it has some very unique features discussed in Section 5.1.

The whole new Section:

 5.1. Features of Lithuanian

 Here are some features of the Lithuanian language that make it interesting and important to include.

The Lithuanian language is highly inflective (fusional) and derivationally complex. Differently from agglutinative languages (that rely on prefixes, suffixes, and infixes), for inflection, Lithuanian "fuses" inflectional categories together, whereas prefixes, suffixes, and infixes are still used to derive words. E.g., a diminutive/hypocoristic word can be derived by adding suffixes to the root, and the word can have two-three suffixes (sometimes going up to six), each added suffix changing its meaning slightly. The language has compounds (connecting two, three words). Besides, verbs can be made from any onomatopoeia; phrasal verbs (e.g., go in, go out) are composed by adding the prefix to the verb. 

Some sentence structures are preferable in the Lithuanian language, but syntactically there is a lot of freedom in composing sentences. However, it is important to notice, that word order changes the sentence shade and things that are emphasized.

All this complexity and variety of forms makes isolated Lithuanian words ambiguous: 47% of Lithuanian word forms are morphologically ambiguous [110]. This in turn makes diacritic restoration and typo correction even more challenging. 

> 6.  Other keyboard layouts besides qwerty could also be considered and analyzed

QWERTZ and AZERTY keyboards that are specific to Croatian, French, Hungarian, and Slovak among the languages used (see Table 1) were also analyzed, as explained at the end of Section 6.2.:

For the Croatian, French, Hungarian, and Slovak languages, correspondingly to their different keyboard layout family (see Table 1) we remapped the original English QWERTY dataset before inferring typo probabilities. E.g., for Croatian, having a QWERTZ layout, we had to swap the letters "z" and "y" when calculating probabilities. In our initial experiments, we did not observe significant model performance differences between the QWERTY and remapped typo generation versions.

> 7.  Provide more explanation on the transformer model ByT5, compared with Bert or Roberta

This is explained in Sections 4.2., 4.2.1, 4.2.3, and 4.2.4. We added an additional practical explanation for our application of the difference at the end of Section 4.2.4.:

The sequence-to-sequence nature of ByT5 tackles the limitations of the latest state-of-the-art diacritics restoration [46] which is based on BERT. The latter system was of an auto-encoding type and performed classification for each token. I.e., it had to predict the proper classes of each token correction, described by the position and diacritic sign type. This system is limited to its predefined instruction set (correction classes), which is highly language-dependent, and the single task of restoring diacritics. On the other hand, our sequence-to-sequence ByT5 approach allows us to address multiple grammatical errors and learn to generate output sequences in a much more universal, language-independent approach.

> 8.  Provide more explanation on data validation used in the experiments

We maintained the same experimental conditions (Wikipedia texts, preprocessing code, train-validation splitting) as used in similar research works to keep results comparable. We provide the links in the Data Availability Statement at the end of the article. The quality of the data was investigated visually to be OK, but that's probably not something important to report in the article.

> 9.  Add explanation on real-world application of this model

We added a paragraph in the Discussion:

All this makes this approach very promising for large-scale real-world applications. Our combined diacritic restoration and typo correction solution could in principle already be used in, for example, auto-correcting text messages or social media posts/comments. Expanding the approach in the ways discussed above opens even bigger application horizons.

Sincerely,

Authors

Reviewer 3 Report

This is an interesting and well-structured article that performs a rather original approach to correct both spelling errors and diacritic marks. I am having below some comments on the presentation:

  • Line 409: Marian NMT, give a reference
  • Lines 760-61 and generally in the article: It is not very clear, how the authors generated the samples for the Lithuanian language. The link https://ufal.mff.cuni.cz/~majlis/w2c/download.html provides corpora for various languages. Additionally, the data at http://hdl.handle.net/11234/1-2607 do not contain Lithuanian. The paper implies on the lines above, that the data were obtained, using the tools at reference [38], which essentially are the same as the first link, above. I initially got the impression that the authors, could somehow, collect text for Lithuanian. As this is rather confusing, please be more specific of what is included.
  • Table 3 legend: “… as well as a single seven times longer training run”. Better change it to “a training approach with seven times more samples (x7)”, as this is what you are actually doing.
  • Line 835: “we report models trained for x7 and x17 times”. How were those numbers chosen? Is it a research practice or is it indicated somewhere in the bibliography?
  • Line 898: I would recommend changing the section 7.1.1 title to “An approach with dictionary and ByT5 (Dict.+ByT5)
  • Line 957: Rephrase “Under-training is most significant for Vietnamese” to “Under-training, has significant impact to the performance, for Vietnamese”.
  • Discussion: I think a comment about the requirements to perform spell-checking and correction to diacritics, would be interesting, for languages that don’t have alphabets that can be somewhat modelled by Latin, such as Cyrillic languages and Arabic.

Author Response

Dear reviewer,

Thank you for your time and positive review.

> This is an interesting and well-structured article that performs a rather original approach to correct both spelling errors and diacritic marks. I am having below some comments on the presentation:

Thank you.

> Line 409: Marian NMT, give a reference

Reference added.

> Lines 760-61 and generally in the article: It is not very clear, how the authors generated the samples for the Lithuanian language. The link https://ufal.mff.cuni.cz/~majlis/w2c/download.html provides corpora for various languages. Additionally, the data at http://hdl.handle.net/11234/1-2607 do not contain Lithuanian. The paper implies on the lines above, that the data were obtained, using the tools at reference [38], which essentially are the same as the first link, above. I initially got the impression that the authors, could somehow, collect text for Lithuanian. As this is rather confusing, please be more specific of what is included.

A very good remark. We now explicitly added (in Data Availability Statement), that data for 12 benchmark languages were used as it is from http://hdl.handle.net/11234/1-2607 while data for Lithuanian were taken from https://ufal.mff.cuni.cz/~majlis/w2c/download.html and preprocessed with tools from https://github.com/arahusky/diacritics_restoration. This is exactly the same data source and the same procedure the authors from [38] did to prepare data for mentioned 12 other languages.

> Table 3 legend: “… as well as a single seven times longer training run”. Better change it to “a training approach with seven times more samples (x7)”, as this is what you are actually doing.

You are right that this refers to a number of samples, but not training time (which is almost, but not quite the same in our case). However, we don't want to make a false impression that we are training the model with seven times more data. The amount of training data available is in any case limited by the size of Wikipedia of the corresponding language (Table 1). We make this now more explicit in 3rd paragraph of Section 6.1.

We rephrased the legend to sound "... for three separate training runs with different initial model weights and dataset samplings trained for 524 288 sentences (#samples: ×1) and a single run for eight times more (×8), cycling through the available training data (Table 1) as needed.", to make this clear.

> Line 835: “we report models trained for x7 and x17 times”. How were those numbers chosen? Is it a research practice or is it indicated somewhere in the bibliography? 

The research practice is to report on 2^x steps, as is done in the T5 paper [100]. Yet in our case, we, unfortunately, chose the complex change of batch size from 256 to 8192 during the training at 6000 steps. This combined with the aim to train as long as possible with the available modest hardware resulted in those ceiling-rounded numbers, that are now reported as x8 and x19 to be more precise. We added an explanation for this in Section 6.1.

> Line 898: I would recommend changing the section 7.1.1 title to “An approach with dictionary and ByT5 (Dict.+ByT5)

Changed as suggested.

> Line 957: Rephrase “Under-training is most significant for Vietnamese” to “Under-training, has significant impact to the performance, for Vietnamese”.

Rephrased to "The lack of training hurts the performance of the Vietnamese the most, ..."

> Discussion: I think a comment about the requirements to perform spell-checking and correction to diacritics, would be interesting, for languages that don’t have alphabets that can be somewhat modelled by Latin, such as Cyrillic languages and Arabic.

We added a paragraph in the Discussion:

Our approach is also easy to scale to other languages, as it does not depend on the alphabet or structure of the language. For example, only the typo dataset generation model in this work depends on the Latin alphabet and a corresponding keyboard layout.

Sincerely,

Authors

Round 2

Reviewer 2 Report

The authors already revised according to the reviewer suggestions. Please check all the grammatical error before resubmit the manuscript.